# Landslide Susceptibility Assessment Model Construction Using Typical Machine Learning for the Three Gorges Reservoir Area in China

**Junying Cheng** [1] , **Xiaoai Dai** [1,*] , **Zekun Wang** [2], **Jingzhong Li** [3] , **Ge Qu** [1] , **Weile Li** [4] , **Jinxing She** [1] and **Youlin Wang** [5]

1   School of Earth Science, Chengdu University of Technology, Chengdu 610059, China;
    chunjunying@stu.cdut.edu.cn (J.C.); 2021020025@stu.cdut.edu.cn (G.Q.); sjx@stu.cdut.edu.cn (J.S.)
2   Department of Mechanical Engineering, Auburn University, Auburn, AL 36849, USA; zzw0043@auburn.edu
3   School of Resource and Environment Sciences, Wuhan University, Wuhan 430072, China;
    00009232@whu.edu.cn
4   State Key Laboratory of Geohazard Prevention and Geoenvironment Protection,
    Chengdu University of Technology, Chengdu 610059, China; liweile08@mail.cdut.edu.cn
5   Northwest Engineering Corporation Limited, Xi'an 710065, China; wangyoulin@nwh.cn
*   Correspondence: daixiaoa@mail.cdut.edu.cn; Tel.: +86-180-8198-8616

**Abstract:** The Three Gorges Reservoir region in China is the Yangtze River Economic Zone's natural treasure trove. Its natural environment has an important role in development. The unique and fragile ecosystem in the Yangtze River's Three Gorges Reservoir region is prone to natural disasters, including soil erosion, landslides, debris flows, landslides, and earthquakes. Therefore, to better alleviate these threats, an accurate and comprehensive assessment of the susceptibility of this area is required. In this study, based on the collection of relevant data and existing research results, we applied machine learning models, including logistic regression (LR), the random forest model (RF), and the support vector machine (SVM) model, to analyze landslide susceptibility in the Yangtze River's Three Gorges Reservoir region to analyze landslide events in the whole study region. The models identified five categories (i.e., topographic, geological, ecological, meteorological, and human engineering activities), with nine independent variables, influencing landslide susceptibility. The accuracy of landslide susceptibility derived from different models and raster cells was then verified by the accuracy, recall, F1-score, ROC curve, and AUC of each model. The results illustrate that the accuracy of different machine learning algorithms is ranked as SVM > RF > LR. The LR model has the lowest generalization ability. The SVM model performs well in all regions of the study area, with an AUC value of 0.9708 for the entire Three Gorges Reservoir area, indicating that the SVM model possesses a strong spatial generalization ability as well as the highest robustness and can be adapted as a real-time model for assessing regional landslide susceptibility.

**Keywords:** landslide; spatial modeling; spatial generalization ability; support vector machine; ROC curve

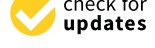



## 1. Introduction

Landslide is a destructive geological phenomenon globally, with a wide distribution, high frequency, and solid destructive power, influenced by groundwater and surface water under the action of gravity; and large rock masses on the slope and the entire sliding surface undergo the process of sliding [1]. It is more hazardous in areas with substantial topographic relief and slope [2–4]. China is in the eastern region of Asia, with complicated geological formations and a massive mountainous terrain. China is one of the most severely threatened countries globally by landslides. Faced with various types of landslide disasters every year, the southwest region of China is particularly notable. Landslides cause irreversible and immeasurable losses to multiple aspects of construction and development

and people's lives and properties in China [5,6]. Meanwhile, secondary disasters caused by landslides can also block rivers, triggering floods, and even cause the formation of mudslide disasters, with more severe losses [7,8].

Scholars have researched landslide susceptibility assessment models. They have made time-sensitive progress when establishing that evaluation models are crucial for regional landslide susceptibility assessment. Conventional models can be classified as qualitative, quantitative, and semi-quantitative evaluation models. Qualitative and semi-quantitative evaluation models are collectively known as knowledge-driven models. The knowledge-driven models mainly export system scoring methods and hierarchical analysis [9]. Quantitative evaluation models, also known as data-driven models, are used for landslide prediction by statistically analyzing evaluation factor data to reveal the intrinsic association between each factor and landslide occurrence. Standard data-driven models consist of binary-statistics-based information volume models [10–13], the entropy index model [10–13], the frequency ratio model [14–16], the logistic regression model based on multiple regression statistics [17–21], artificial neural network models based on machine learning [17–21], the support vector machine model [22–25], the decision tree model [22–25], random forest [26–35], neuro-fuzzy [36–40], fuzzy [41–45], and so on. Orahan Osman et al. performed landslide sensitivity mapping for the Arhavi-Kabisre river basin, where they used five machine learning techniques, ANN, LR, SVM, RF, and CART, to produce a landslide inventory of 252 landslide events in the basin while selecting 11 landslide adjustment factors such as altitude, slope, plane curvature, soil, lithology, distance from road, distance from river, and TWI by the ReliefF method to make a landslide sensitivity map. Finally, ROC, sensitivity, specificity, F-measure, accuracy, and Kappa index are applied to compare and validate the performance of the five machine learning techniques. It is concluded that the artificial neural network model has the highest predictive ability for landslide sensitivity mapping in the region [46]. Yu Lanbing et al. addressed the large number of landslides that occur in the Three Gorges Reservoir area of the Yangtze River due to periodic regulation of reservoir water levels. Using the Wushan section as the study object, 165 landslides were identified in this section and 14 landslide impact factors were selected from different data source constructions using multicollinearity analysis and IGR modeling methods. The computational results of four machine learning models, SVM, ANN, CART, and LR, were used for landslide sensitivity mapping. The accuracy of these four maps was also evaluated using ROC and accuracy statistics. The results show that the SVM model performs best in this study and can be used for sensitivity modeling in the Three Gorges Reservoir area and other landslide-prone areas [47]. Xie Wei et al. proposed that landslide sensitivity mapping (LSM) may be an effective method for landslide hazard prevention and damage mitigation. In their research, they developed a hybrid approach, including GeoDetector and machine learning clusters, to provide a new perspective on how to address landslide hazard prevention and loss mitigation. The machine learning cluster consists of four models, ANN, BN, LR, and SVM, and they will automatically select the best model to generate LSM. The four models are evaluated using the ROC curve, prediction accuracy, and the seed cell area index (SCAI) methods. The results showed that the SVM model performed the best in the machine learning cluster in an area under the ROC curve of 0.928 and an accuracy of 83.86%. Thus, the support vector machine mapped the landslide sensitivity of the study area to coincide with the landslide inventory, indicating that the hybrid approach is effective in screening landslide impacts and assessing landslide sensitivity [48]. Pei Xiangjun et al. used 1022 seismic landslide sites in Jiuzhaigou National Geopark, the hardest-hit area of the 8 August 2017, *Mw*6.5 earthquake, as sample data for regional evaluation of landslide susceptibility. They selected 16 landslide control and influence factors, such as seismic parameters, topography, geological conditions, hydrological conditions, and human engineering activities, using LR models in slope units under different factor combinations, and 30 evaluation models were established. Finally, the data were sampled by 10-fold cross validation and the model's accuracy and prediction accuracy were evaluated using ROC curve models. The results show that the LR model has good applicability in evaluat-

ing earthquake and landslide susceptibility areas in the Jiuzhaigou region, and it is also concluded that the model has stability robustness [49].

Machine learning has gradually become the core of artificial intelligence research and one of the fastest-growing disciplines of artificial intelligence, with a wide range of applications, thanks to the rapid development of computer technology. The earliest machine learning algorithms date back to the early 20th century. After decades of progress, multiple classical methods have been invented. Machine learning is now widely used in geological catastrophe research. The analysis results of "machine learning" and "landslide" or "algorithm" and "landslide" through "Web of Science" show that since 2010, in all, 2658 types of research have been performed on landslide hazards and machine learning or algorithms by domestic and foreign scholars. The result of "machine learning" and "landslide susceptibility" or "algorithm "and "landslide susceptibility" is 1241 items, as shown in Figure 1.

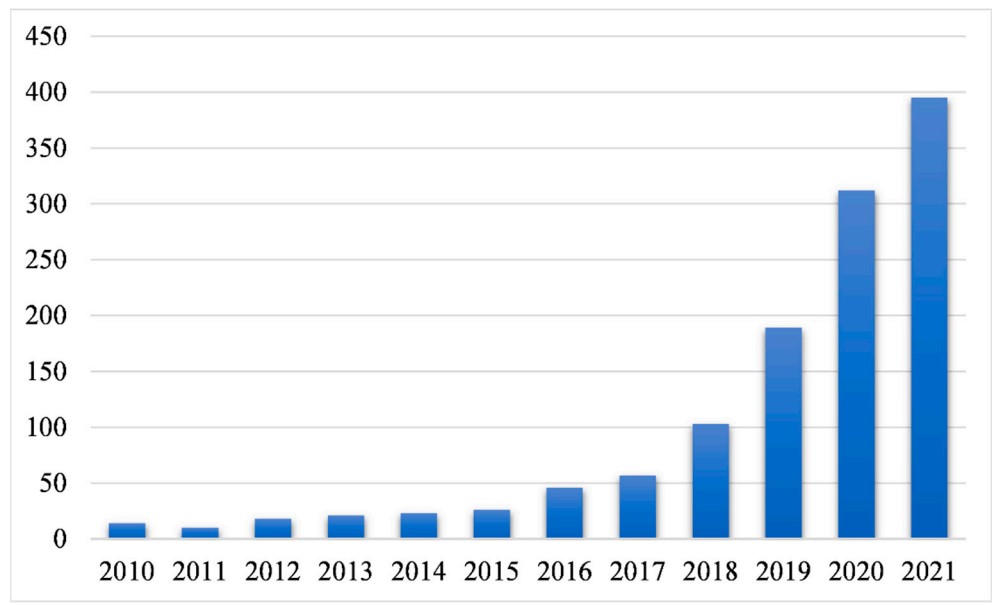

**Figure 1.** Statistics on the number of machine learning publications in landslide susceptibility studies.

The above data visualization results show that according to the statistical graph of the number of articles published, the research articles on landslide susceptibility and machine learning have increased yearly since 2016. The number of articles published in the last two years has increased sharply, indicating that machine learning has been increasingly researched and applied in landslide susceptibility in recent years. Machine learning models are extensively employed in the assessment of landslide susceptibility across the world, and they are more successful in addressing the present often-occurring geological catastrophe study area. The major goal of the research was to apply machine learning to find a landslide susceptibility assessment method that was appropriate for the study location [50].

Landslide susceptibility may be assessed using a variety of methods. Traditional statistical analysis methods are mostly used to assess and analyze the susceptibility of landslides, according to the results of a "web of science" literature search. Various machine learning approaches, such as logistic regression, artificial neural networks, support vector machines, decision trees, random forests, and other models, are increasingly used to assess landslide hazard susceptibility. Machine learning approaches are now more efficient than classical statistical methods and heuristic models [51,52]. Furthermore, every little improvement in model accuracy will have a more significant influence on landslide susceptibility outcomes. As a result, comparative research of machine learning approaches is required to obtain accurate landslide susceptibility assessment results.

The primary deficiencies in the current research are as follows: (1) In the existing machine learning evaluation models for landslide susceptibility, there is a lack of exploration of the spatial generalization ability of the models. (2) There is less research on landslide susceptibility in terms of the interaction validation of the indicator factors, leading to deficiency in the accuracy rate of the landslide hazard research results. In response to the previous research, this paper presents an evaluation analysis of landslide susceptibility in the Three Gorges Reservoir area of the Yangtze River, China. This research field is used to compare and evaluate the performance of logistic regression, random forest, and support vector machine model methods.

## 2. Materials

### 2.1. Study Area

The Three Gorges Reservoir Dam on the Yangtze River, an important water conservancy project in China, and the Three Gorges Reservoir area are a significant ecological barrier in China, provide abundant water for irrigation in the Yangtze River basin and have a considerable role in economic prosperity along the Yangtze River, promoting the economic development of the western region and balancing the East–West differences. The Three Gorges Reservoir region extends between 28°30′N and 31°45′N latitudes and 105°50′E and 111°42′E longitudes. It is connected with the Sichuan Basin, covering four districts and counties under the jurisdiction of Yichang City in Hubei Province and 22 districts and counties under the jurisdiction of Chongqing City, with a total area of about 79,000 square kilometers and submerged arable land of 19,400 hectares. The Three Gorges Reservoir area ecosystem is unique and fragile, with frequent natural disasters. At the same time, the Three Gorges Reservoir area is an ecological treasure trove for the Yangtze River Economic Belt and the whole country.

The Three Gorges Reservoir area of the Yangtze River is located in the mid-latitude subtropical monsoon climate zone, influenced by alternating winter and summer winds. The temperature and precipitation change significantly in seasons. The climatic characteristics are undeniably due to the complex terrain. According to the annual monitoring data, there is less precipitation along the river valley in the study area and the average annual rainfall increases by about 55 mm for every 100 m increase in elevation. The rainy season is from May to September every year, and the rainfall during this period accounts for 70% and more of the rainfall in the whole year, and there are many heavy rainstorms. The climate of the Three Gorges Reservoir area, with abundant rainfall and heavy rainfall, is one of the main triggering factors for landslide geological disasters in the reservoir area.

The study area is located in the transition zone from the second to the third terrace of China's topography and is the junction of the east Sichuan fold and the west Hubei mountains, with a middle and low mountain erosion canyon landscape. The east-west part of the reservoir area traverses two natural geographic units, roughly bounded by Fengjie, with the eastern part is the Three Gorges Canyon deeply embedded in the Wushan Mountains and the western part is the low mountainous hilly area in the eastern part of the Sichuan Basin. The loose rock pore water in the study area is mainly stored in the loose accumulation layer and slope accumulation layer of the Quaternary system. It is primarily recharged by precipitation, fracture water of the underlying bedrock, or karst water, which is influenced by seasonal changes. The dynamic instability of groundwater level is one of the main factors affecting the stability of landslides in the area.

Geological hazards in the study area are widely distributed, numerous, large scale, and serious. Landslides are the most prominent geological hazard in the reservoir area. With a large number of developments, landslides are large scale and strong. At the same time, with the rapid development of social economy, the scale and intensity of human engineering activities in this area have continued to expand, their impact on the natural environment has become increasingly serious, and they have become one of the important triggering factors of geological hazards in the area, mainly in the construction of migrant towns, reservoir construction, deforestation, mining, and so on. These activities adversely

affect the rock and soil bodies near the slopes, destroy the natural ecosystem, cause soil erosion, and seriously damage the original natural morphological structure and stress balance, which are important causes of landslides and collapse disasters. The geographical location of the study area is shown in Figure 2.

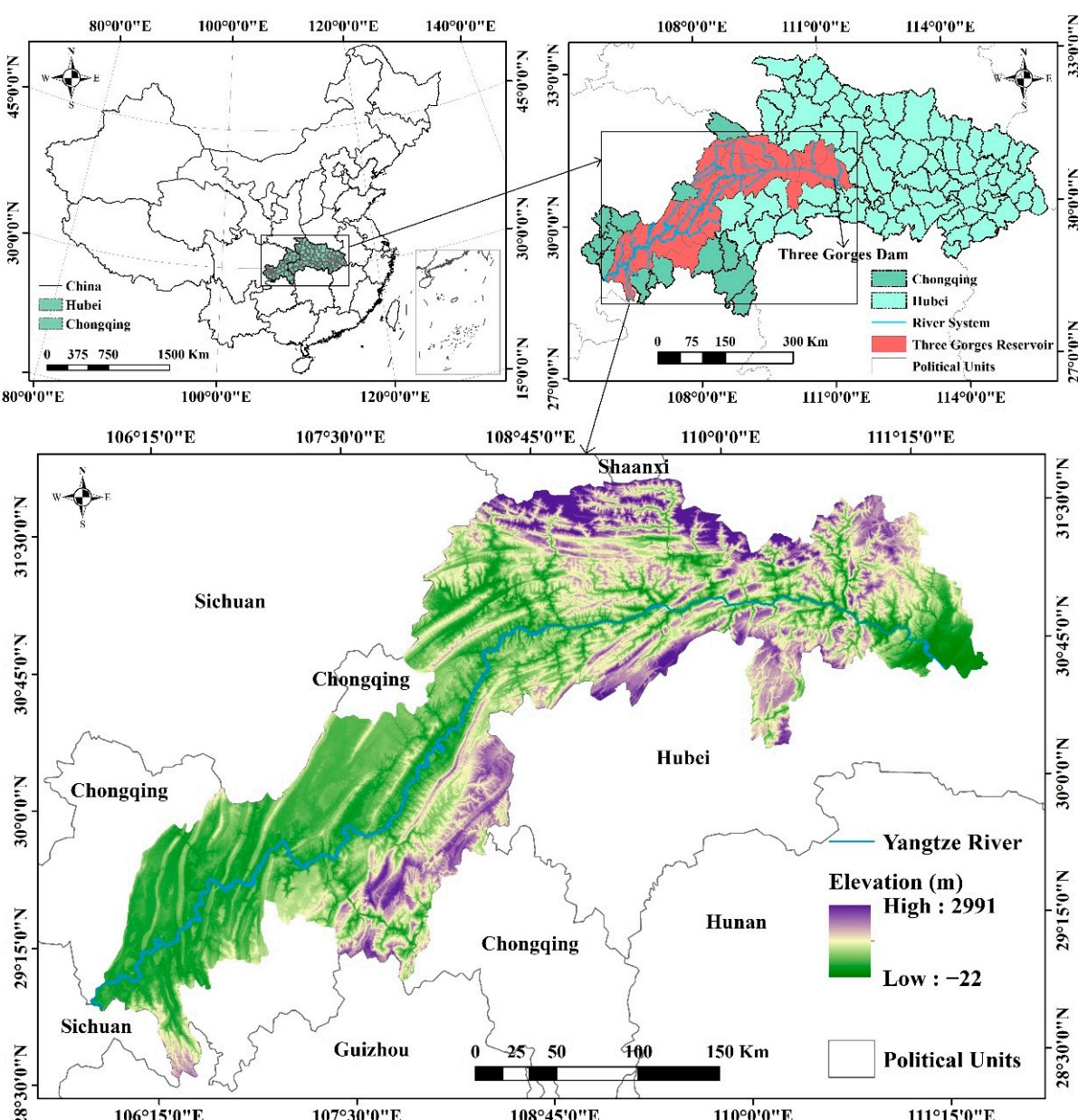

**Figure 2.** Geographical location of the Three Gorges Reservoir Area.

### 2.2. Database

Historical landslide hazard data used in this paper were obtained from the Institute of Geographic Sciences and Natural Resources Research, Chinese Academy of Sciences (http://www.resdc.cn/ (accessed on 27 April 2021)). Based on geological hazard survey information combined with remote sensing images, landslide data points were derived through remote sensing visual interpretation to establish a spatial database of landslides in the study area with an accuracy of 30 m. This spatial database of landslides includes two main parts: (1) historical landslide hazard datasets in the Three Gorges Reservoir area up to 2019 and (2) a dataset of indicators that affect landslide susceptibility.

### 2.2.1. Landslide Dataset

Landslide datasets are essential for investigating and analyzing regional landslide hazards and risks. Following reservoir storage, landslides and debris flows have increased due to the complicated geological conditions and disasters in the Three Gorges Reservoir area. Landslides in the Three Gorges Reservoir area mainly include accumulation layer landslides, bedding rock landslides, dangerous rock mass landslides, unstable slopes, and reservoir banks. The external factors affecting the deformation of geological disasters in the reservoir area mainly include reservoir water, rainfall, and human engineering activities. In the initial stage of the Three Gorges Reservoir impoundment, reservoir water was the main inducing factor for the deformation geological disasters in the Three Gorges Reservoir area. Rainfall has become the dominant trigger during recent years' high watermark operation. Under the action of different external forces, different levels of landslide disasters occur every year in this area. There are 9539 landslides in the landslide cataloging data, including 3661 large landslide events, 3852 medium landslide events, 1432 small landslide events, and 594 other types of landslide events. The spatial distribution of landslide sites in the Three Gorges Reservoir area is shown in Figure 3.

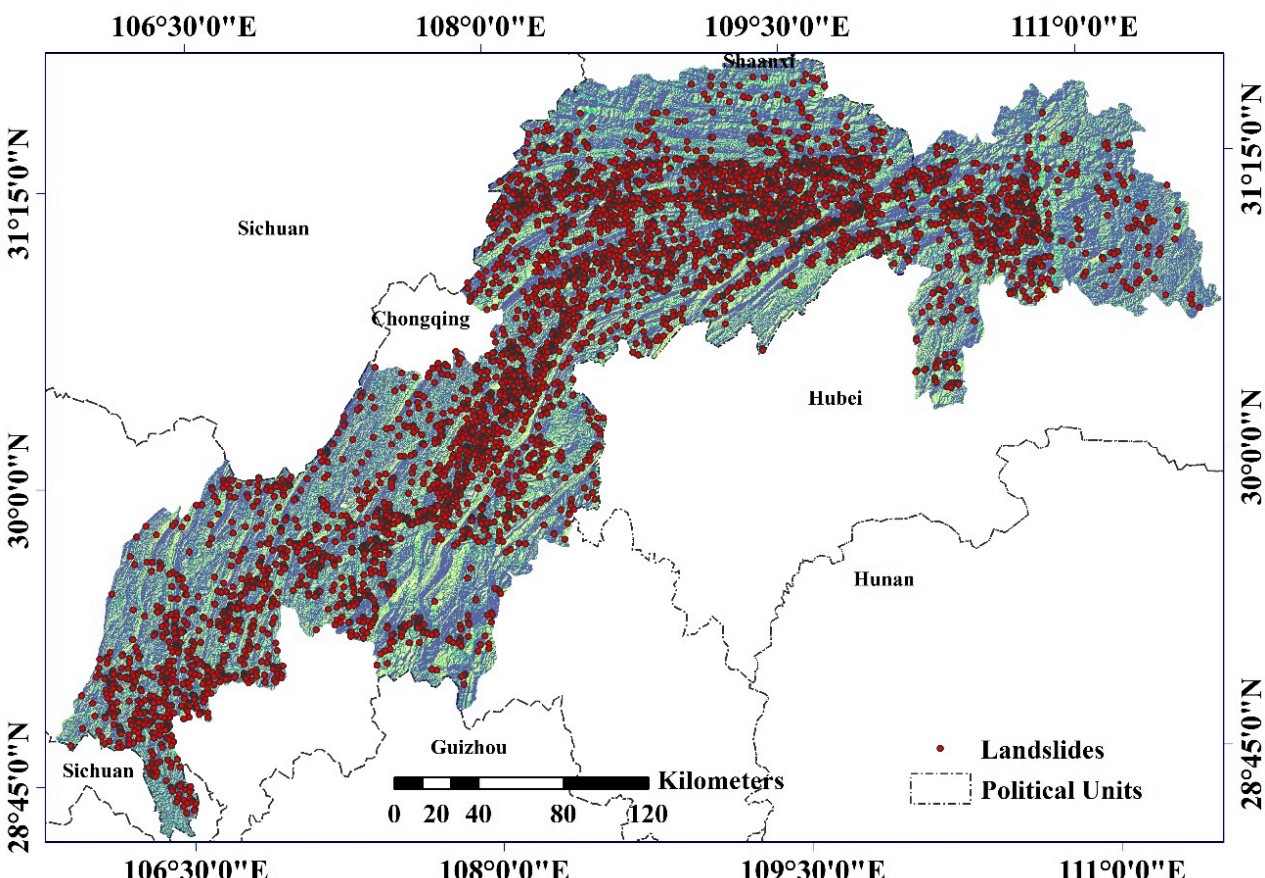

**Figure 3.** The spatial distribution of landslide points in the Three Gorges Reservoir area.

### 2.2.2. Assessment Indicator Data of Landslide Susceptibility

Landslide susceptibility assessment index data are topography, geological, land cover, ecological, meteorological, seismic, and human engineering activities. In this study, 14 index factors of topographic, geological, ecological, meteorological, and human engineering activities are selected for the research and analysis of landslide susceptibility assessment in the Three Gorges Reservoir region, with the goal of better understanding the environment and reservoir ecology in the study area.

Table 1 below describes the data sources and the extracted individual indicator factors for this paper, with a uniform resolution of 90 m for all data and a consistent projection coordinate system Krasovsky_1940_Albers.

**Table 1.** Data source and causes of selection.

| Data Type | Factors | Causes of Selecting the Parameters |
|---|---|---|
| Topographic | Elevation | Landslides are more probable on high slopes than on low slopes in mountainous landscapes [53]. |
| | Slope | Slope is crucial in preventing landslides in a given location. The susceptibility to landslides is very high in areas with steep slopes [54]. |
| | Terrain Ruggedness Index (TRI) | The TRI reflects changes in surface relief and erosion [55]. |
| | Slope Length (LS) | Different slope length indicators have varying degrees of influence on landslide incidence [54]. |
| | Curvature | The chances of landslides decrease as the curvature value decreases [56]. |
| | Plan Curvature | The amount of divergence and convergence in the direction of water flow on the slope determines the plan curvature. It impacts the rate at which water flows downward, erosion and deposition processes, and consequently the occurrence of landslides. |
| | Profile Curvature | Profile curvature directs the water flow in the slope, converging or diverging, and impacts the landslide sliding. |
| | Topographic Wetness Index (TWI) | TWI indicates the moisture content of the soil. Areas with high humidity and suitable conditions are prone to landslides [57,58]. |
| Geological | Distance to Fault | The frequency of landslides is relatively high around extremely active fracture zones [59]. |
| | Lithology | Landslide sensitivity varies based on the lithological character of the area [60]. |
| | Distance to River | The more distant the area from the river, the less frequent the landslides [61]. |
| Ecological | Normalized Difference Vegetation Index (NDVI) | It directly affects the degree of slope transformation and soil erosion [62]. |
| Meteorological | Precipitation | Precipitation is one of the critical triggering factors that induce landslides. Heavy precipitation effectively separates soil and rock, increasing the risk of landslides [53]. |
| Human Engineering Activity | Distance to Road | Road construction makes the side slopes unstable and is one of the primary triggers of landslides [63] |

## 3. Method

### 3.1. Methodology

To construct an evaluation model for landslide susceptibility, the relationship between indicator factors and landslides was analyzed for historical landslide hazard data, topography, geological, ecological, land cover, meteorological and human engineering activities, and other influential data. During the study, nine indicator factors with high mutual independence were screened by Pearson correlation coefficient and covariance diagnosis. The historical landslide hazard data were divided into 70% training samples and 30% validation samples. The indicator data were validated by establishing RF, LR, and SVM models and the 10-fold cross-validation method. Then, the accuracy was evaluated for the global area and the local area of the study area. Finally, a landslide susceptibility assessment map was drawn, with a discussion of the factors that influence landslides in the area.

The main research ideas are as follows (Figure 4):

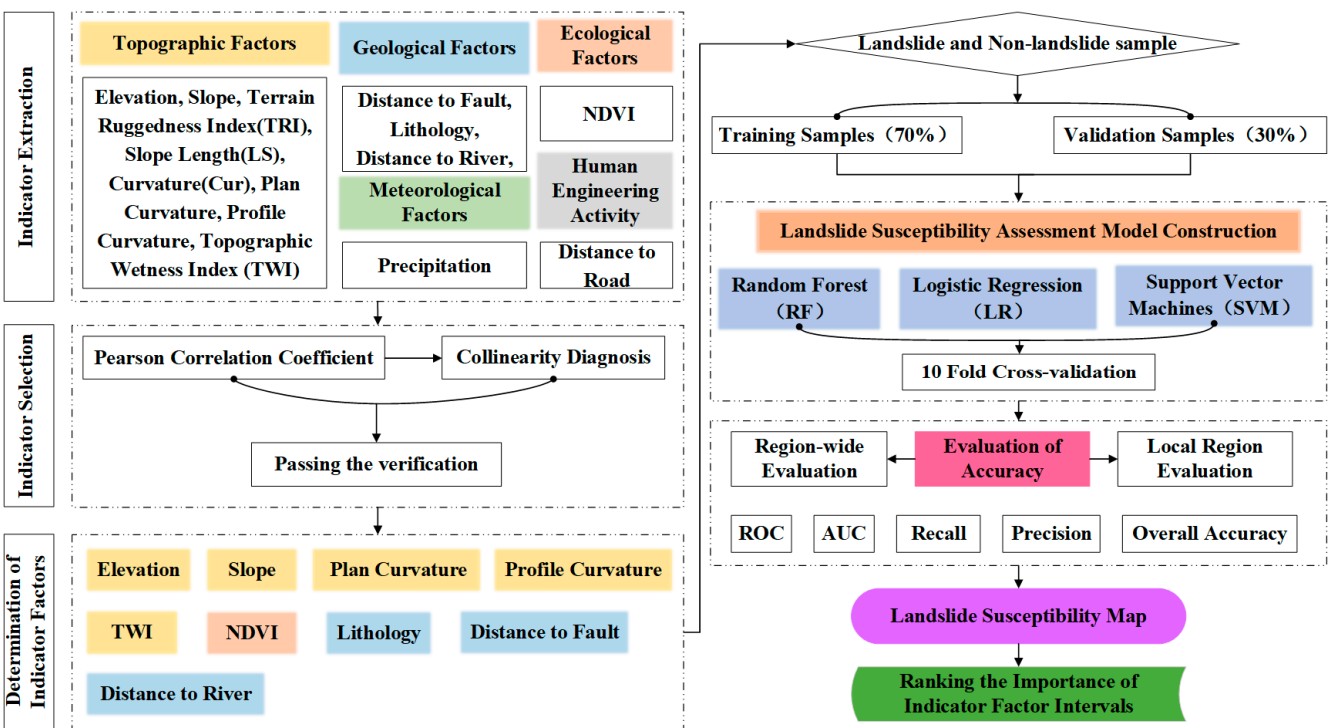

**Figure 4.** Flowchart showing the methodology of the present work.

### *3.2. Assessment Indicator Screening Methods*

#### 3.2.1. Pearson Correlation Coefficient

The landslide susceptibility assessment model results depend on the data quality of the selected indicator factors because the redundancy of data in the modeling process will lower the model's predictive ability [51]. Therefore, it is essential to screen the optimal indicator factors of the landslide susceptibility assessment model. The correlation coefficient can measure the linear relationship between the landslide indicator factors [64]. In this study, the indicator factors are screened by The Pearson correlation coefficient (*PCC*) and eliminated with high correlation. The *PCC* is a metric for determining the degree of linear correlation between two variables with values ranging from −1 to 1. The closer the correlation coefficient's absolute value is to 1, the more similar the sample is; the closer it is to 0, the less similar it is. *PCC* > 0 indicates a positive correlation between the two factors, while *PCC* < 0 indicates a negative correlation. It is usually considered that there is a solid correlation between the variables when the correlation coefficient is between 0.8 and 1.0; 0.6–0.8 indicates a strong correlation, 0.4–0.6 indicates a moderate correlation, 0.2–0.4 is a weak correlation, and 0.0–0.2 is a very weak or no correlation. For the two sets of samples, the *PCC* between them can be expressed by Equation (1) as:

$$PCC = \frac{\sum\limits_{i=1}^{n}(x_i - \overline{x})\sum\limits_{j=1}^{n}(y_i - \overline{y})}{\sqrt{\sum\limits_{i=1}^{n}(x_i - \overline{x})^2\sum\limits_{j=1}^{n}(y_i - \overline{y})^2}} \tag{1}$$

where *PCC* denotes the correlation coefficient between samples $x_i$ and $y_j$, $x_i$ and $y_j$ denote the variable values of $X_i$ and $Y_j$, and $\overline{x}$ and $\overline{y}$ represent the average value of $X_i$ and $Y_j$, respectively. The more significant the absolute value of the *PCC*, the greater the correlation between the indicator factors affecting landslide hazards [65].

### 3.2.2. Extraction of Indicator Factors

The correlations between the indicator factors were analyzed by the Pearson correlation coefficient in the study. The factor combinations with strong correlations derived from the analysis results were combined and the combinations with *PCC* > 0.7 filtered out. A correlation greater than 0.7 means a strong correlation between two factors, so one of the factors should be discarded [66]. After screening, nine indicator factors were identified: elevation, slope, plan curvature, profile curvature, distance to fault, lithology, topographic wetness index, NDVI, and distance to river. The results are shown in Figure 5.

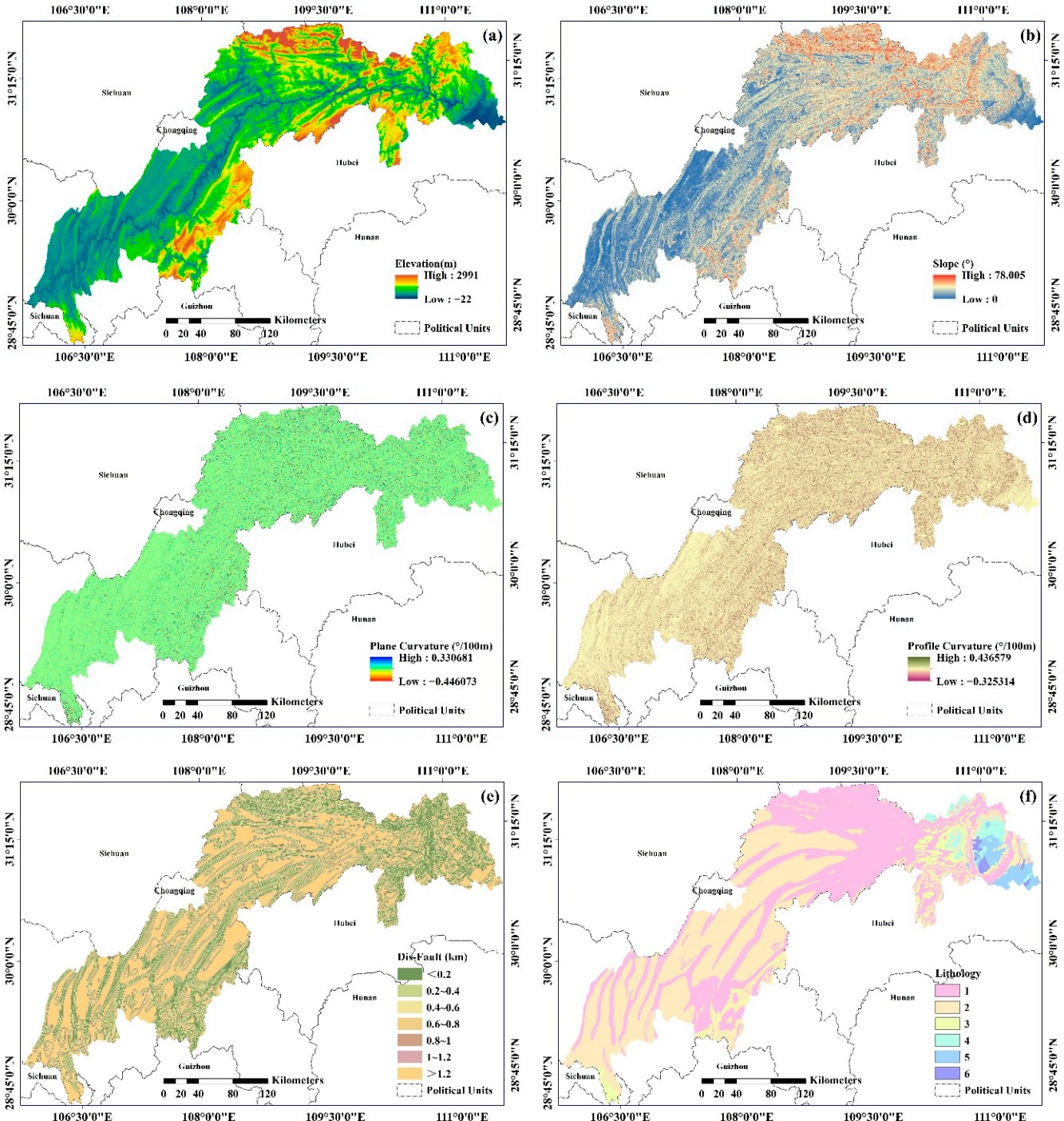

**Figure 5.** *Cont.*

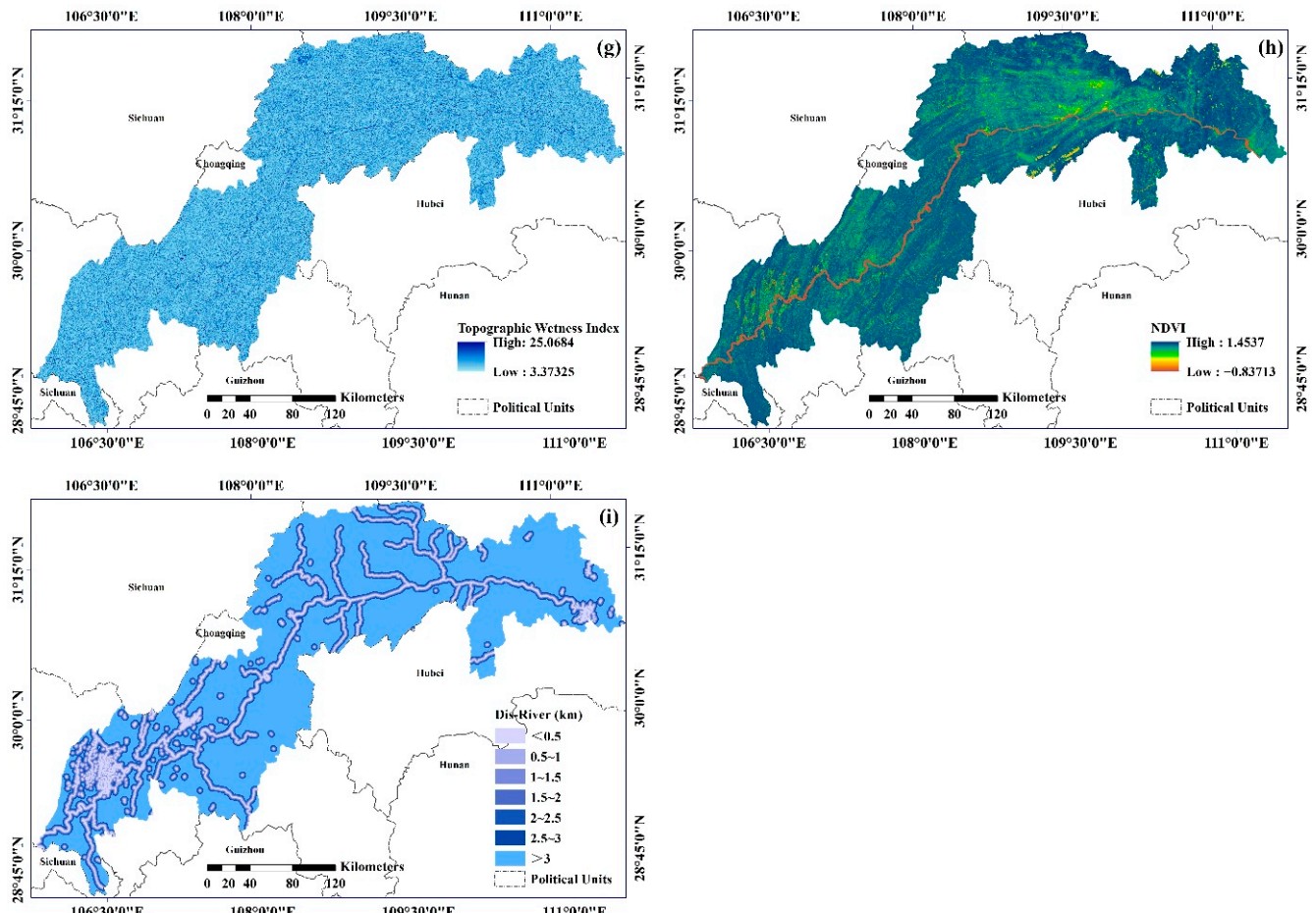

**Figure 5.** Indicator factor maps used for landslide susceptibility modeling. (**a**) Elevation, (**b**) slope, (**c**) plan curvature, (**d**) profile curvature, (**e**) distance to fault, (**f**) lithology, (**g**) TWI, (**h**) NDVI, (**i**) and distance to rivers.

3.2.3. Landslide Frequency of Different Indicator Factors

After screening 9 indicator factors from 14 indicator factors with high correlation affecting landslide susceptibility in this study area by the Pearson correlation coefficient, it is necessary to investigate the association between landslides and indicator factors. Frequency statistics examined the association between landslides and non-landslides for each indicator factor. First, the cumulative frequency values in the range of 5%~95% were analyzed and the effect of extreme values in the study was excluded, from which the number of landslide events and the frequency of landslide occurrence in different indicator factor range intervals were obtained, as shown in Figure 6. Figure 6a shows the distribution of landslides in different elevation ranges. The landslide frequency tends to decrease and increase and decrease as the elevation value increases, indicating that the landslide susceptibility interval is within a particular elevation range. Figure 6b shows the change in landslide number in different slope ranges. With the slope increasing, the frequency first increases, then decreases, and then increases; in the slope interval more significant than 50°, the landslide frequency reaches the highest value, indicating that a landslide is most likely to occur. Figure 6c shows the changes in the number of landslides in different intervals of the plane curvature. From the statistical chart, it can be seen that the overall decline trend is intuitive. The frequency is −0.02~0.02, and the number of landslides is the main effect. The profile curvature in Figure 6d shows a similar variation to the plane curvature, from which it can be seen that the density of landslide events is higher in the area where the profile curvature is in the range of −0.03°/100 m~0.03°/100 m. Among them, the density of regional landslide events in the range of −0.03°/100 m~0°/100 m

is the largest which indicates that most landslides are concentrated in relatively gentle concave or convex areas. However, because the section curvature is gentle in the interval of 0~0.01°/100 m and tends to be a straight slope, the curvature changes in the range of −0.03~0°/100 m and 0.01~0.03°/100 m are more obvious, which are manifested as convex slopes and concave slopes, so the frequency of landslides in this range is relatively high. Figure 6e shows the variation in landslide events at different distances to fault. As the distance to fault decreases, the frequency of landslides decreases and as the distance to fault increases, the frequency of landslides increases. When the fault distance exceeds 1200 m, the frequency of landslides reaches the highest value, which indicates that landslides are most likely to occur during this interval. Figure 6f shows an increasing trend in the frequency of landslides in the TWI, most of which occur in less than 7, indicating that landslides are mainly concentrated in the interval smaller TWI values. For the NDVI in Figure 6g, the frequency of landslides generally increases first and then decreases, with landslides occurring mainly in a larger interval. Figure 6h shows the distance to river. For this factor, the overall trend falls, similar to the distance to fault. The occurrence of landslides is mainly concentrated in long-distance intervals. Since lithology is a categorical variable based on the geological age, continuous frequency distribution statistics are not available and landslides have no trend corresponding to different types of lithology, so this study does not discuss the relationship between lithology and landslides. There is a linear relationship between different indicator factors and the frequency of landslides, which indicates that the factors selected in the study have different effects on landslide occurrence.

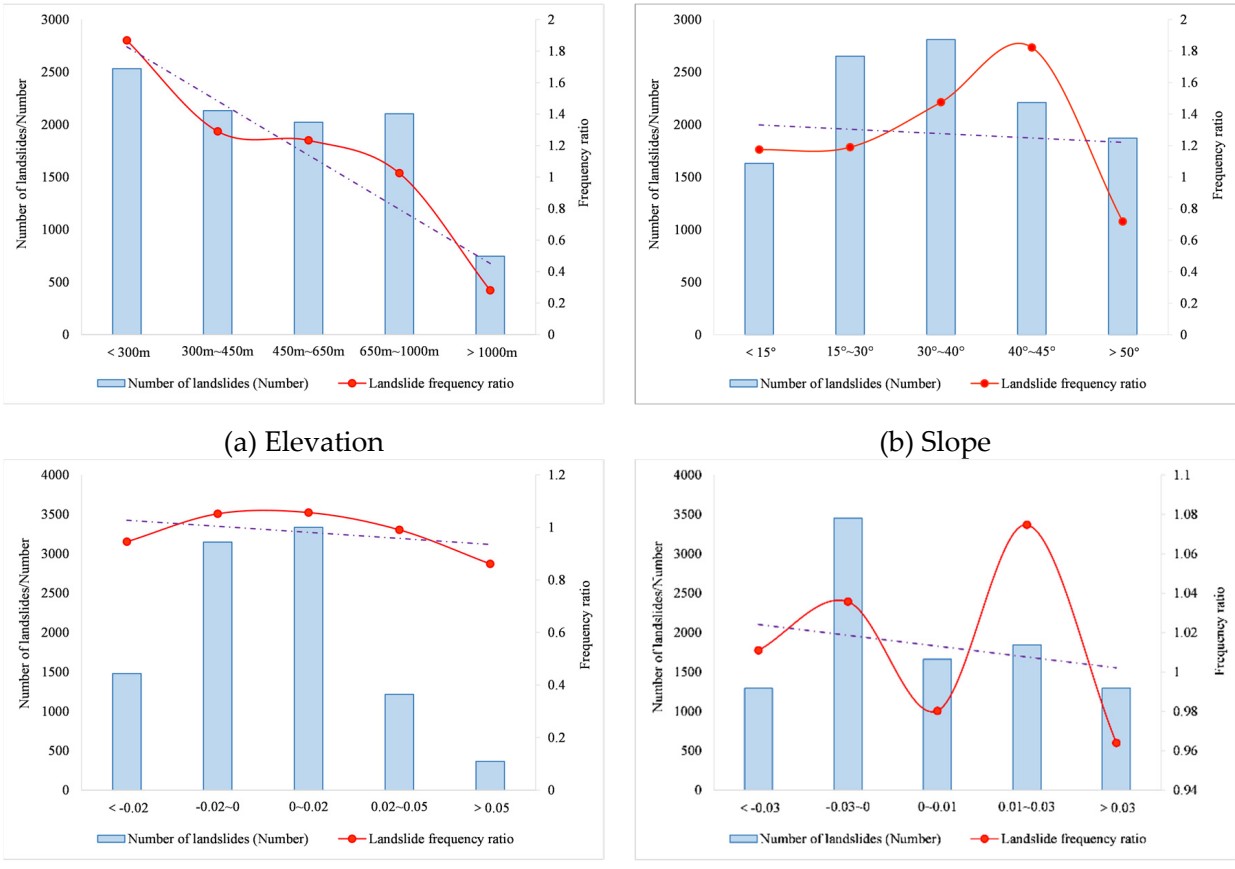

(a) Elevation

(b) Slope

(c) Plane curvature

(d) Profile curvature

**Figure 6.** *Cont.*

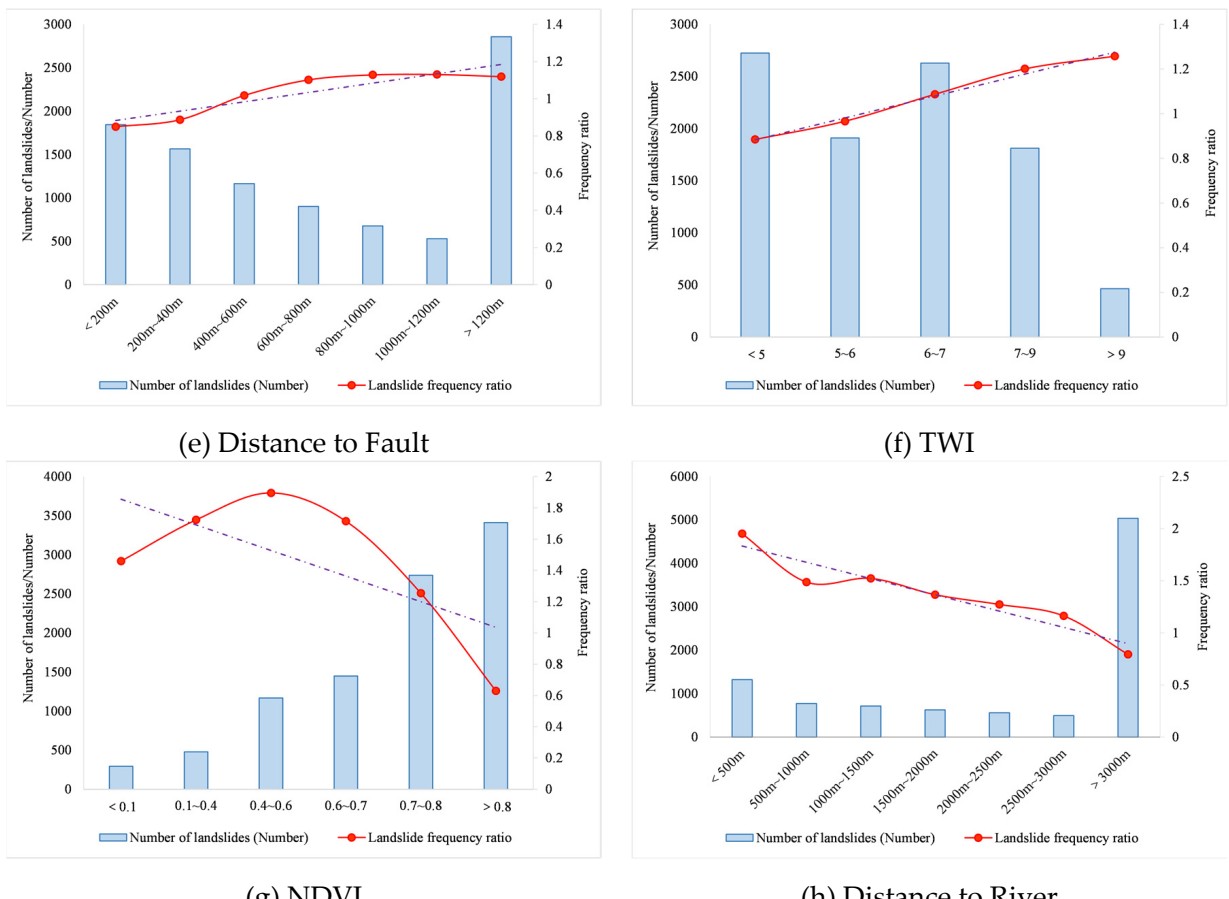

**Figure 6.** Landslide histogram of different indicator factors.

### 3.3. Machine Learning Models

This study uses the logistic regression, random forest, and support vector machine algorithms in the database to evaluate and process historical landslide data from the specified Three Gorges Reservoir area, using the machine learning database's built-in Python language. The outcomes of each model algorithm's analysis are compared based on the algorithm structure of various models.

### 3.3.1. Logistic Regression

Logistic regression (LR) is a standard linear regression analysis model for establishing the relationship between constraints and landslides [67,68]. Regression problems are based on categorical variables, dealing with linear relationships between numerical interval variables, based on a set of predictor variables that predict the probability of an event occurring with a binary variable (such as 0 and 1 or true and false) [69]. The model's independent variables are independent factors, and the best fitting function is determined to characterize the relationship between landslide occurrence and causes quantitatively. Equation (2) in LR expresses the link between the chance of landslides and the independent variable:

$$p = \frac{1}{1 + exp(-z)} \tag{2}$$

where $p$ denotes landslide probability and $z$ is a weighted linear combination of independent variables. Generally, LR uses the following equation (Equation (3)) to fit the dependent variable:

$$Y = Logit(p) = ln(p/1 - p) = C_0 + C_1 X_1 + C_2 X_2 + \cdots + C_n X_n \tag{3}$$

The probability that the dependent variable ($Y$) is 1 is denoted by $p$. $p/(1-p)$ is the likelihood ratio of its sample. $C_0$ is the equation's constant and $C_1, C_2, \cdots C_n$ are the coefficients used to measure the contribution of an independent factor $(X_1, X_2, \cdots, X_n)$ to the change in the dependent variable $Y$.

### 3.3.2. Random Forest

Random forest (RF) belongs to the category of ensemble learning algorithm of Bagging theory, a classification technique using the CART decision tree as the base classifiers, and was first proposed by Breiman [70], who proposed that RF provides a generalization error limit value. The trees that make up a random forest can be either classification trees or regression trees. Each node in a decision tree is segmented using the best feature to generate the best solution from all the features [71]. A training set is first created by RF using the bootstrap method [70]. Then, a decision tree is built for each training set, with each decision tree acting as a classifier. Using each training subset to train different classifiers, integrate all classification results, and assign the subset category with the most votes as the final prediction output (Equation (4)).

$$H(X) = av_k max_Y \sum_{i=1}^{k} I(h_i(X) = Y) \tag{4}$$

$H(X)$ denotes the combined classification model, $h_i$ is a decision tree, $Y$ denotes the output variable, and $I$ denotes the feature function. The marginal function can be expressed through Equation (5) as:

$$mg(X, Y) = av_k I(h_k(X) = Y) - max_{j \neq Y} av_k I(h_k(Y) = j) \tag{5}$$

The more significant the function value, the greater the model's classification reliability. The following is a summary of the categorization (Equation (6)):

$$PE^* = P_{XY}(mg(X, Y) < 0) \tag{6}$$

where $(X, Y)$ is the probability space. As the number of decision trees increases, all sequences will change (Equation (7)) to:

$$P_{xy}(P_\theta(h(X, \theta) = Y) - max P_\theta(h(X, \theta) = j)) < 0 \tag{7}$$

The main advantage of RF is resistance to over-training and the development of sizeable RF numbers, which will not constitute the risk of overfitting, while not requiring scaling, transformations, or changes in algorithm parameters. For predictors, RF is resistant to outliers and automatically manages missing values [72]. In this paper, the Gini coefficient error is reduced by computing the correlation of the attribute subset at each node of the sample, iterating its gradual convergence, and thus computing using the Gini criterion to find the best node split [70]. These criteria measure the degree of correlation between variables and results. The Gini criteria (Equations (8) and (9)) are expressed as:

$$Gini(k, x_i) = \sum_{i=1}^{m} \frac{a_i}{n_s} I(k_{ui}) \tag{8}$$

$$I(k_{ui}) = 1 - \sum_{i=0}^{c} \frac{n_{ci}^2}{a_i^2} \tag{9}$$

The number of landslides $k$ at each node is denoted by $m$, and the number of training input feature vectors is denoted by $n_s$. The distribution of class labels on nodes is $I(k_{ui})$. The value $p$ is the feature variable $x_i \in X$ at node $k$, where $x_i = \{u_1, u_2, \cdots u_m\}$ is the number of samples of value $u_i$ at node $k$, $n_{ci}$ is the sample of $c_i$ pertaining to $u_i$, and $a_i$ is the number of samples of value $u_i$ at node $k$.

### 3.3.3. Support Vector Machine

Support vector machine (SVM) is a standard nonlinear supervised classification machine learning algorithm that Vapnik first proposed [73,74]. SVM is used to discriminate between different forms of training data by searching an N-dimensional hyperplane [75,76]. To discover the optimal separation hyperplane, the core theory leverages the training data to turn the input space into a high-dimensional feature space using the inner product function. In any classification interval, the distance between the hyperplane and the nearest training data point is maximized [77]. Existing research has demonstrated that SVM with the maximal edge classifier has the best generalization ability to invisible data [73].

Assume the sample, where $x_i \in R^n$, $y_i \in \{+1, -1\}$, where $i$ = 1, 2, 3, ..., $m$, and $y_i$ denotes the number of training samples to classify landslides and non-landslides. In the scenarios in this paper, $x$ is an input spatial vector including elevation, slope, plan curvature, profile curvature, distance to fault, lithology, topographic wetness index, NDVI, and distance to river; 1 and 0 indicate landslides and non-landslides. Linearly separable sample data, introduce a slack variable $\xi_i$, and add a penalty term C > 0 for the slack variable to the separation hyperplane objective function, and the optimal separation hyperplane function becomes (Equations (10) and (11)):

$$Min\left(\frac{1}{2} \parallel \vec{w} \parallel^2 + C\sum_{i=1}^{n} \xi_i\right) \tag{10}$$

$$s.t. y_i\left(\vec{w} \cdot \vec{x}_i + b\right) - 1 + \xi_i \geq 0 \tag{11}$$

where $w$ is the weight vector that controls the trade-off between the complexity of the decision function and the number of disqualified training samples, $b$ is the offset, $\xi_i$ is the positive slack variable that allows data points that violate the penalty constraint, and $C$ is the penalty parameter that controls the trade-off between the complexity of the decision function and the number of disqualified training samples. The optimal hyperplane is determined using Lagrange multipliers to solve the following optimization problems (Equations (12) and (13)) [78]:

$$Max\left(\sum_i a_i - \frac{1}{2}\sum_{ij} a_i a_j y_i y_j \left(\vec{x}_i \vec{x}_j\right)\right) \tag{12}$$

$$\sum_i a_i y_i = 0, \ 0 \leq a_i \leq C \tag{13}$$

where $a_i$ is the Lagrangian multiplier and $C$ is the penalty factor. The decision function used to classify the new data (Equations (14) and (15)) is:

$$g(x) = sgn\left(\sum_{i=1}^{n} y_i a_i K(x_i, x_j) + b\right) \tag{14}$$

$$K(x_i, x_j) = e^{-\gamma(x_i - x_j)^2} \tag{15}$$

$K(x_i, x_j)$ denotes the kernel function. To study the ideal hyperplane, the RBF is employed as the kernel function of the SVM model in this paper. When compared to the most widely used functions nowadays, RBF has a nonlinear solid mapping capacity and may be used to partition landslide susceptibility in various ways [79].

### 3.4. Model Accuracy Assessment Criteria

### 3.4.1. Accuracy Analysis

Cross validation can be used to check how well evaluation models predict outcomes. In the process of landslide susceptibility evaluation, historical landslide events are allocated according to Training samples:Validation samples = 7:3 and the landslide event data are

cross validated. The precision, recall, and F1-score statistical indexes are used to compare and assess the performance of the landslide susceptibility model. In addition, the true category of sample data and the predicted category of the learner are analyzed through the dichotomous classification problem using a confusion matrix.

### 3.4.2. Receiver Operating Characteristic

The predictive ability of landslide susceptibility assessment results directly or indirectly affects the local control of landslide hazards. In this study, to predict and compare the model's performance for landslide susceptibility assessment, the Receiver Operating Characteristic (ROC) was introduced to analyze the model's accuracy. ROC is one of the effective methods for characterizing the quality of 1-Specificity and susceptibility detection, particularly in landslide susceptibility assessment. It expresses the model's accuracy through the overall accuracy and predicts the quality of a system by expressing its capacity to correctly predict the occurrence or non-occurrence of events [80].

### 4. Results

#### *4.1. Landslide Susceptibility Assessment Maps for Different Models*

Landslide susceptibility assessment maps describe the quantitative relationship between known landslides and indicator factors and combine theoretical predictions with practical mitigation measures. The LR, RF, and SVM models of the following machine learning methods are used to map the distribution of landslide susceptibility in the research area. The study area is graded based on previous experience and statistical results combined with the results of its grid cell analysis. Among them, the susceptibility is divided into five grades. The susceptibility value of the very high susceptibility area is in the range of 0–0.06, the susceptibility value of the high susceptibility area is in the range of 0.06–0.11, and the susceptibility value of the moderate susceptibility area is in the range of 0.11–0.45, the susceptibility value of the low susceptibility zone is in the range of 0.45–0.88, and the susceptibility value of the very low susceptibility is in the range of 0.88–1.0, as shown in Figure 7 below.

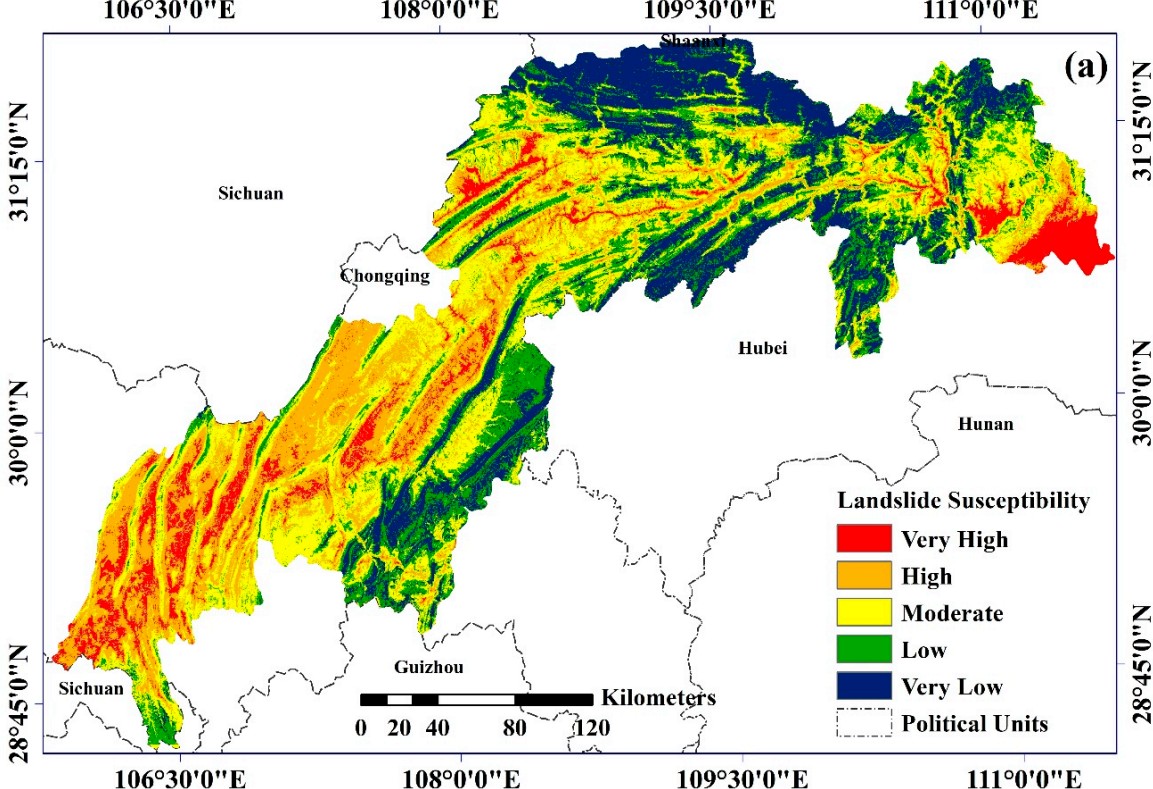

**Figure 7.** *Cont.*

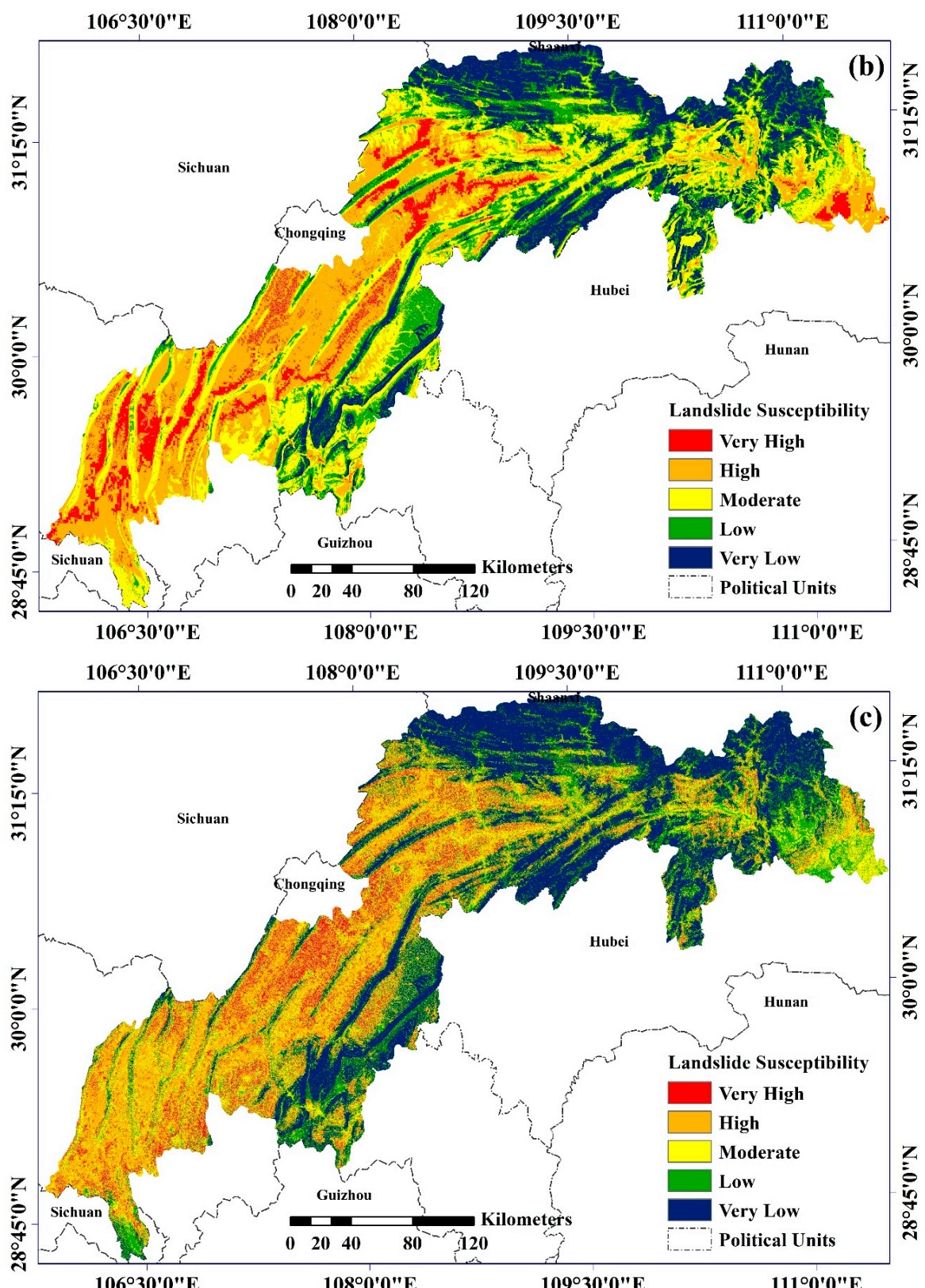

**Figure 7.** Assessment of landslide susceptibility: (**a**) Logistic regression; (**b**) random forest; (**c**) support vector machine.

The landslide susceptibility of the study area is evaluated using three machine learning models, and the distribution of landslide susceptibility in each class is presented in Table 2.

Each model's landslide susceptibility distribution varied greatly, with the LR and RF models predicting similar landslide susceptibility distributions, with the percentage of zones with high and moderate susceptibility occupying almost equal amounts. However, the prediction accuracy of the SVM model differed from the first two models. The high-susceptibility zones occupy 69.17%, the very high-susceptibility zones occupy 12.35%, the moderate-susceptibility zones 10.90%, the low-susceptibility zones 5.41%, and the very low-susceptibility zones occupy 2.17%.

**Table 2.** Percentage of different model landslide susceptibility grades (%).

| Landslide Susceptibility Grade | LR | RF | SVM |
|---|---|---|---|
| Very High | 8.42 | 12.09 | 12.35 |
| High | 37.03 | 35.55 | 69.17 |
| Moderate | 38.01 | 37.26 | 10.90 |
| Low | 11.43 | 12.21 | 5.41 |
| Very Low | 5.12 | 2.89 | 2.17 |

*4.2. Model Accuracy Evaluation*

4.2.1. Accuracy Verification Parameter Evaluation

LR, RF, and SVM institutions in machine learning algorithms were used to create landslide susceptibility assessment models. Precision, recall, F1-score, susceptibility, 1-specificity, and overall accuracy were used to evaluate the model's accuracy and the analysis results. Table 3 shows the results, which are displayed in Figure 8. According to the research results, SVM > RF > LR, i.e., the SVM has the highest accuracy, with 92.23%, and the LR has an accuracy of 81.49% among the three models. In terms of recall, SVM > RF > LR; the ranking is the same as that for precision. Recall of SVM is 92.66% accurate, which is the highest among the three, and RF and LR have similar recall values. F1-score is a mixed metric of precision and recall, with SVM > RF > LR, and the F1-score of SVM is 92.44%. Analyzed from the model's overall accuracy, the SVM has the highest overall accuracy, of 92.43%, compared to the other two models, suggesting that the model checks are highly accurate. The SVM performs best in both overall accuracy and recall. The precision, recall, F1-score, susceptibility, 1-specificity, and overall accuracy are the same for the models. The maximum accuracy of the SVM analysis in this research of the Three Gorges Reservoir area indicates that it has a good prediction potential.

**Table 3.** Comparison of model accuracy.

|  | LR | RF | SVM |
|---|---|---|---|
| Precision (%) | 81.49 | 86.60 | 92.23 |
| Recall (%) | 84.79 | 86.38 | 92.66 |
| F1-score (%) | 83.11 | 86.49 | 92.44 |
| Susceptibility (%) | 84.79 | 86.38 | 92.66 |
| 1-Specificity (%) | 80.75 | 86.58 | 92.20 |

4.2.2. Comparison of ROC and AUC Results

The accuracy of the constructed machine learning model is verified and evaluated through the ROC and the AUC pair. Figure 9 depicts the model comparison results. The results reveal that the machine learning model has a high prediction accuracy in investigating landslide susceptibility in the Three Gorges Reservoir area, based on the ROC and the AUC of each model. The AUC value of the SVM model is 0.9708, which is a higher AUC value compared to the other two models, followed by the RF model's AUC value of 0.9409 and the LR model's AUC value 0.9075. In this research, the ROC is used to verify the performance of the three models, and Figure 9 indicates that the SVM outperforms the other models. The results reveal that the models for landslide susceptibility in the Three Gorges Reservoir area built using machine learning techniques have excellent prediction accuracy, indicating that the SVM is the best model for the Three Gorges Reservoir area study.

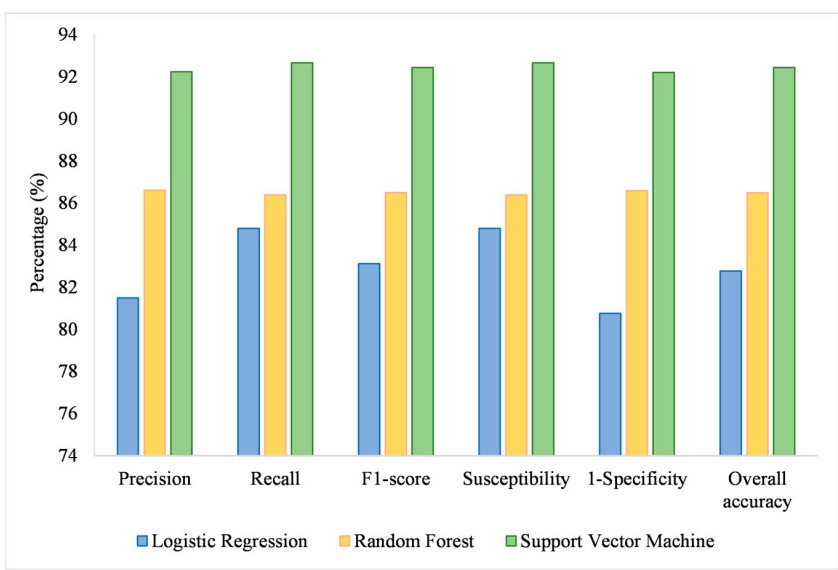

**Figure 8.** Comparison of model accuracy.

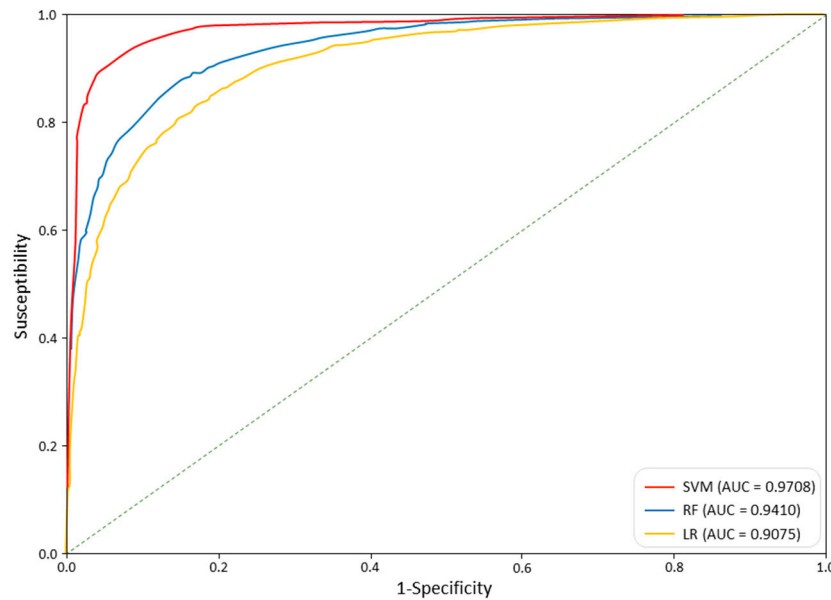

**Figure 9.** Comparison of the three models' ROC curves and AUC values.

From Figure 9, it can be seen that there are some differences in the performance capabilities of precision and overall accuracy of the three different models in the Three Gorges Reservoir area. The figures show that SVM has the best performance in the study area, with an AUC of 0.9708, and RF and LR performed the worst among the three models. It also shows that the SVM established in this paper is a stable and reliable landslide susceptibility assessment model that can be applied to the Three Gorges Reservoir area.

### 4.3. Comparison of Prediction Results of Models

The optimum model SVM produced from the comparison model analysis is used to estimate the research area's landslide susceptibility. The model's accuracy was validated using 30% of the validation samples. When compared to NASA's worldwide landslide susceptibility zoning map, the findings of the training and validation samples are quite close. The research area's accuracy is comparable. To assess the correctness of landslide susceptibility zoning, the geological landslide hazard sites that have now been identified should

be grouped into as many high-susceptibility zones as feasible. The study superimposes actual landslide event points on the susceptibility map to explore the map's distribution of landslide susceptibility (Figure 10).

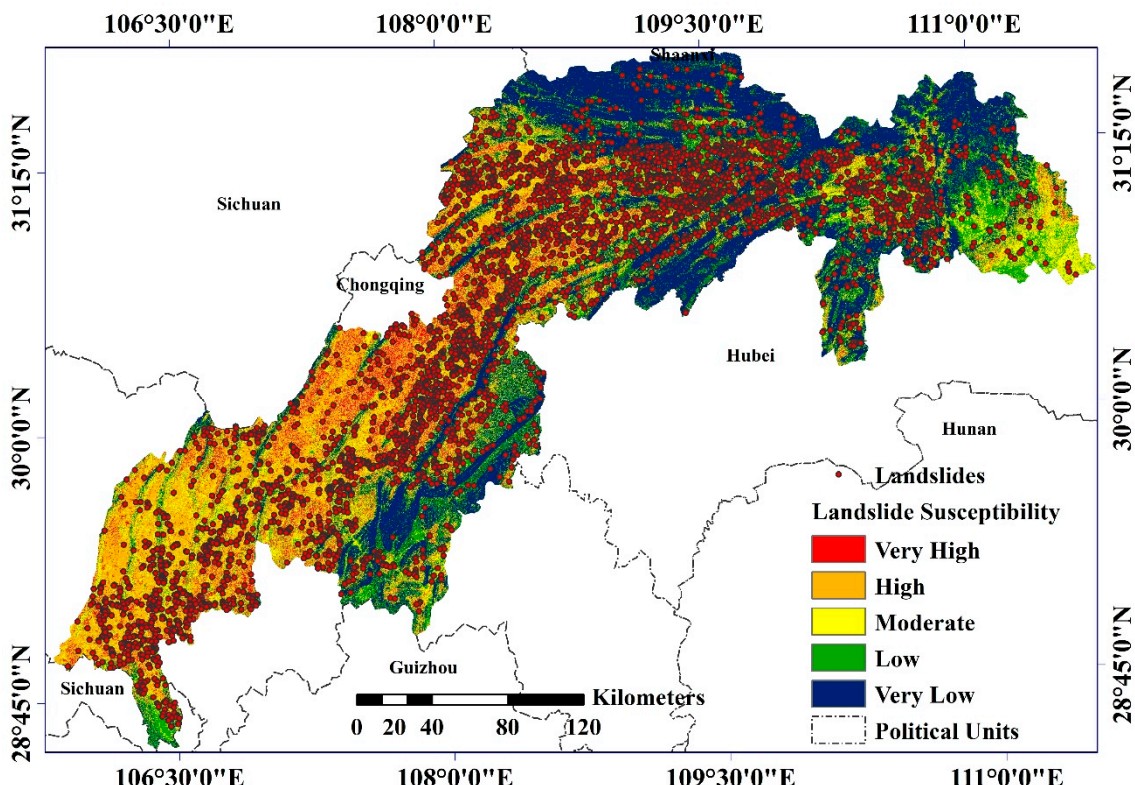

**Figure 10.** Landslide susceptibility map and historical landslide occurrence distribution in the Three Gorges Reservoir area.

The comparison revealed that the known landslides in each region are essentially in a very high-susceptibility zone, with only a few landslides occurring in the areas with low or very low susceptibility. The findings demonstrate the model's correctness and validate the SVM dependability in assessing landslide susceptibility. Since the landslide susceptibility indicator factors identified in this paper include only topographic, geological, ecological, meteorological, and human engineering activity factors, the optimal model SVM obtained by these indicator factors can predict landslide susceptibility in real time.

## 5. Discussion

### 5.1. Comparative Analysis of Local Areas

Landslide susceptibility assessment maps with higher accuracy should show the predictability of new and re-occurring landslides and analyze event data indicating that many existing landslides are in zones of high susceptibility. Figure 11 shows four large landslides near the Three Gorges Reservoir in Yunyang County, Chongqing, at Qingliang Temple, Fanjiayuanzi, Minhe Village, and Ganjiayuanzi Renhe Bridge, all of which are in areas of high landslide susceptibility. However, differences between the three landslide susceptibility assessment maps can be seen. In the landslide susceptibility map derived from the LR, the four landslides in Yunyang County are mainly in areas of high landslide susceptibility, with the surrounding areas being of high and moderate susceptibility. The four landslides in the RF model are mostly in the very high-susceptibility zones, whereas the four landslides in the SVM are mostly in the high- and moderate-susceptibility zones. Figure 11a–c shows three landslide disasters in Yunyang County and Wanzhou District of Chongqing in 2017 and 2021. These landslides are located in the region's high-, moderate-, and low-susceptibility areas.

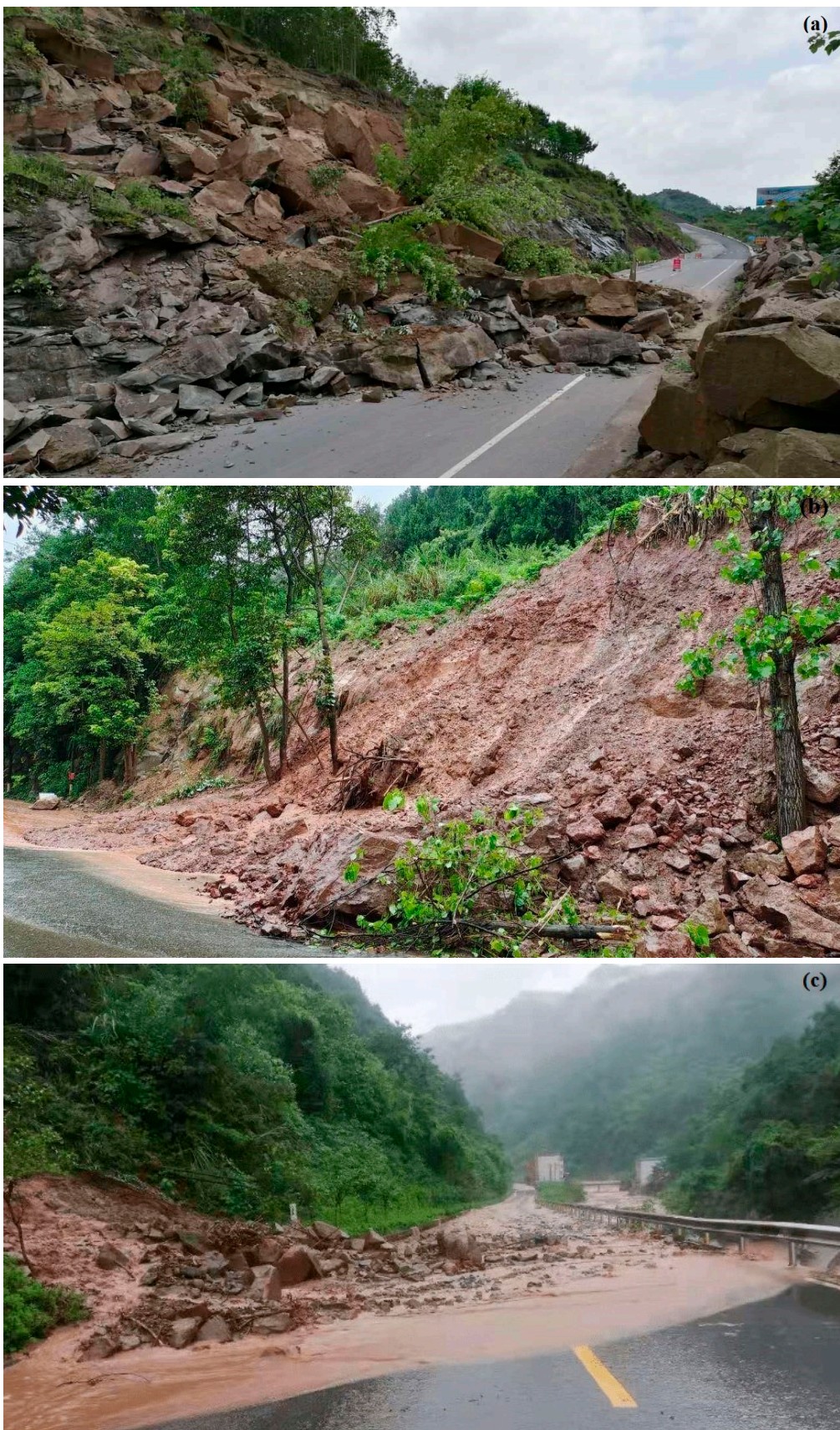

**Figure 11.** *Cont.*

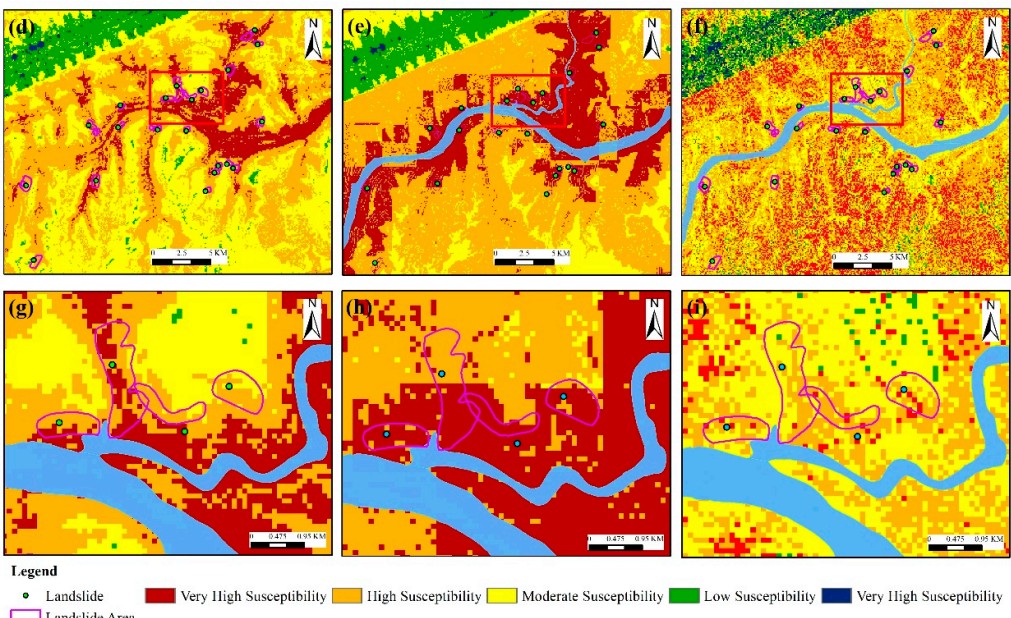

**Figure 11.** Typical landslides in the Three Gorges Reservoir area: (**a**) Yunyang County (high-susceptibility area, June 2017). (**b**) Yunyang Country (moderate-susceptibility area, August 2021). (**c**) Wanzhou District (low-susceptibility area, July 2021). Typical landslide susceptibility analysis results: (**d**) LR analysis results. (**e**) RF analysis results. (**f**) SVM analysis results. Typical local landslide results: (**g**) local results of LR, (**h**) local results of RF, and (**i**) local results of SVM.

## 5.2. Comparing the Spatial Generalization Ability of the Models

ROC curves and AUC values were calculated during the investigation and the percentages of TN, FN, FP, and TP for each model (Figure 12). From the figures, it can be seen that the percentages of TP and TN are more significant for all three models. Still, the percentages are most significant in the SVM model compared to the other two models, thus indicating that the spatial generalization ability of the SVM model is relatively optimal and has better stability within the Three Gorges Reservoir area.

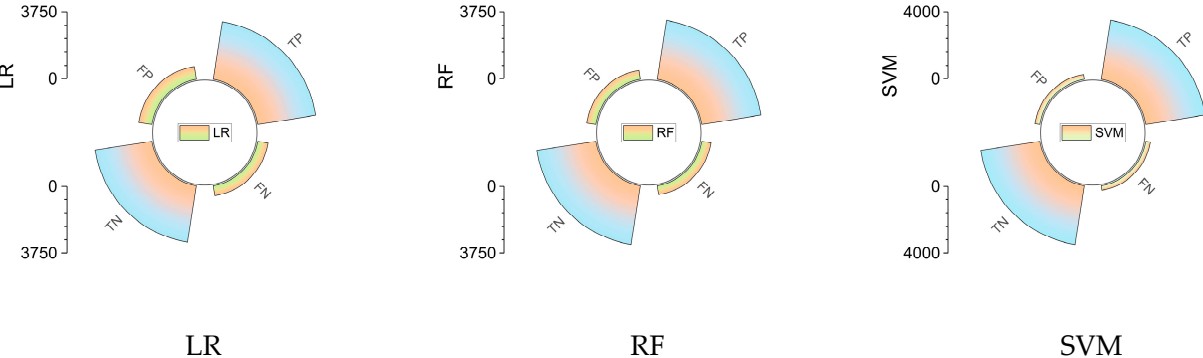

**Figure 12.** Distribution of TN, FN, FP, and TP for different model prediction results.

## 5.3. Support Vector Machine Space Generalization Capability

The SVM is the optimal model in all research areas, as shown by the previous analysis, and the model's application is further appreciated by mastering the SVM's prediction accuracy in the Three Gorges Reservoir study region. Figure 13 shows the distribution of precision, recall, F1-score, and overall accuracy obtained by applying the SVM to the Three Gorges Reservoir area.

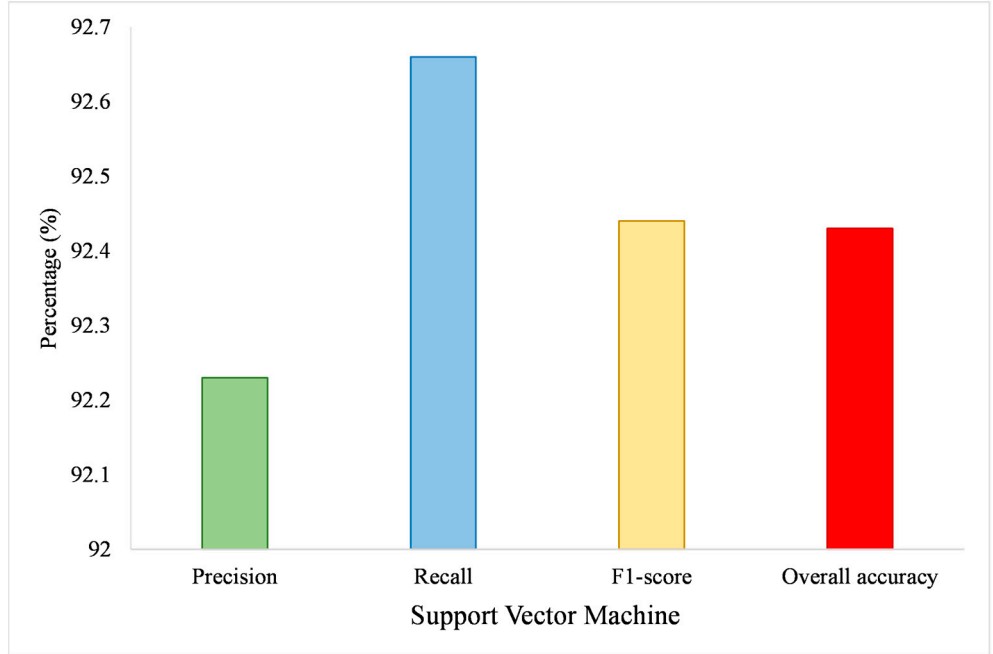

**Figure 13.** Comparison of the prediction accuracy of the support vector machine application in the Three Gorges Reservoir area.

The precision, recall, F1-score, and overall accuracy of the Three Gorges Reservoir area are all over 0.9, as seen in the above figure. When the results are compared, it is clear that SVM has the best accuracy rate in the Three Gorges Reservoir area. The accuracy of different models in the same area varies, and the four indicators can be used to determine the SVM's applicability in the research region. In addition, it is known that the SVM model has strong spatial generalizability in this study area when comparing the distribution of the confusion matrix prediction outcomes of the three models.

Landslides develop in different landslide-prone areas with different patterns, so the susceptibility models perform in varied ways in other areas. This study finds an effective model in the Three Gorges Reservoir area by comparing three machine learning models, LR, RF, and SVM. The results show that the SVM model performs the best. In addition, the performance behavior of SVM for sensitivity modeling in other regions was collected. As shown in the literature in Table 4 below, the accuracy of SVM is always greater than 0.85. We can see that the performance of SVM is acceptable in different regions, so it can be used as a recommended model for the Three Gorges Reservoir area and other landslide-prone areas.

**Table 4.** The accuracy of the SVM model in different areas.

| Authors | Study Area | Accuracy of SVM (%) |
|---|---|---|
| Phong et al. [81] | The Muong Lay district, Vietnam | 87.00 |
| Roy et al. [53] | Darjeeling and Kalimpong Districts, West Bengal, India | 90.00 |
| Xianyu et al. [82] | The Wanzhou of the Three Gorges Area, China | 91.10 |
| Faming et al. [83] | Nantian area in southeastern hilly area, China | 93.17 |
| Bordoni et al. [84] | Oltrepò Pavese (northern Italy) | 97.75 |

*5.4. Ranking of Relative Importance of Factors*

The overall accuracy is used as the rating criterion in the importance ranking of indicator factors. Overall accuracy represents the importance of the classifier for the entire sample judgment modeling process, and this paper specifically refers to the percentage of slippage and non-slippage in the classification process. The importance of the factors in the optimal model SVM of this paper is evaluated. The relative importance of nine

indicator criteria for landslide susceptibility chosen in the study is determined by the model independently rating each factor's weight. As indicated in Figure 14, the factor weights from large to small include distance to river, slope, TWI, profile curvature, distance to fault, plane curvature, lithology, NDVI, and elevation, based on the analytical results of this paper.

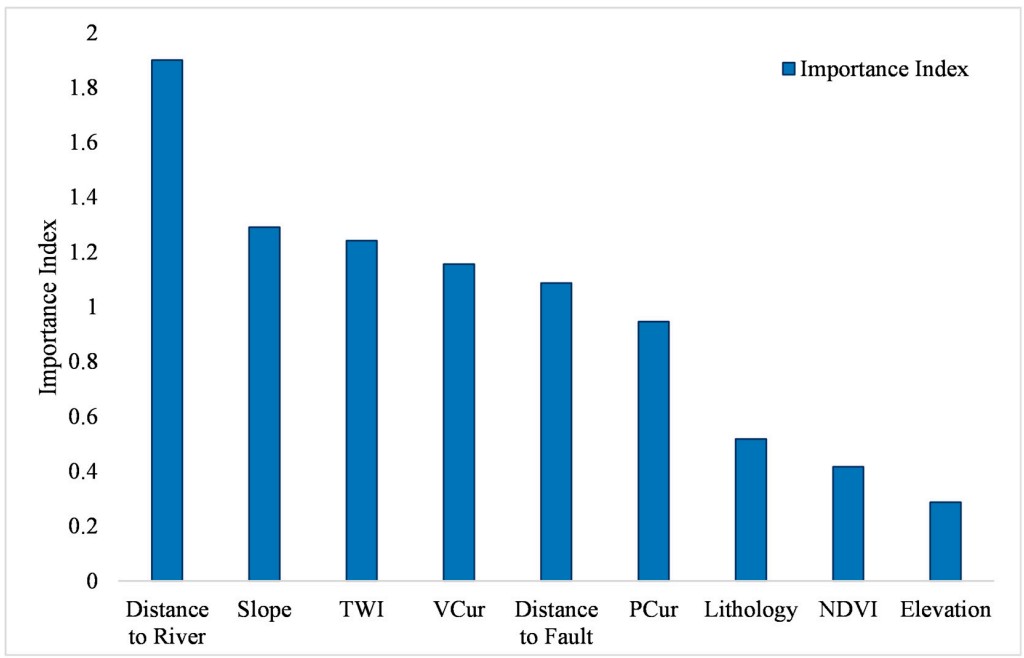

**Figure 14.** Relative importance assessment.

In the Three Gorges Reservoir area, the distance to the river contributes the most to landslides, according to the relevance ranking of the elements in the modeling process. Furthermore, the distance to the river is the most important factor causing landslides and the order of importance is consistent with research in the region [85–87].

### 5.5. Assessment Results and Discussion

The optimal model SVM obtains optimal classification results. The susceptibility index computed by the model was imported into ArcGIS after acquiring the landslide susceptibility map of the Three Gorges Reservoir area. To create the susceptibility assessment map of the study area, the geometric interval approach in ArcGIS was used to divide the susceptibility index into five groups (very low, low, moderate, high, and very high), and the landslide frequency of each class were extracted for statistical analysis by the method of the extract by mask. Figure 15 is obtained.

When the landslide susceptibility assessment map is merged with the landslide hazard events in the Three Gorges Reservoir area, it can be observed that the majority of the landslide hazard events are in areas with high susceptibility. We can learn about landslide occurrence in the study area and forecast landslide geological hazards.

This paper uses landslide hazard incidents in the Three Gorges Reservoir area over the years to build a model. In all, 9539 landslides of various sizes occurred due to the geographic location of the Three Gorges Reservoir area and the establishment of the Three Gorges Dam project, accounting for 75.8% of the geological hazard events in the entire Three Gorges Reservoir area. The goal was to build a model using machine learning methods, examine the benefits of the SVM in this study area, and show that the SVM you built has significant spatial generalization capacity and good robustness and performance. In the Three Gorges Reservoir area, a real-time stability model was used to assess the susceptibility to landslides.

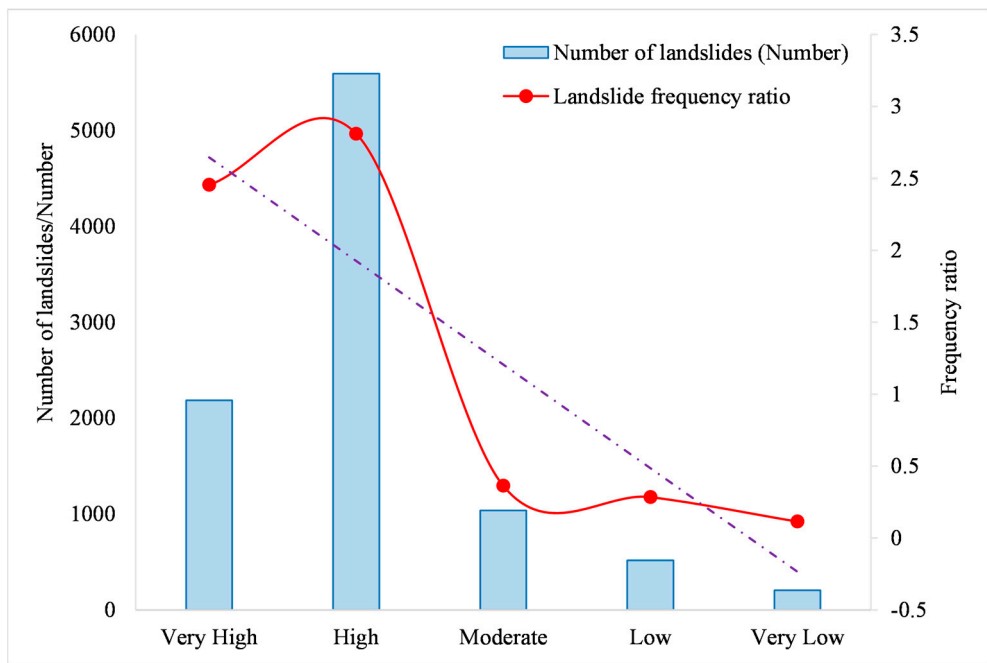

**Figure 15.** Regional statistics of landslide susceptibility.

The Three Gorges Project is located in a critical location, and its huge scale has a profound impact on the development of China. It is the largest water conservancy pivot project in China's and even the world's history, with considerable benefits in flood control, electricity generation, and shipping, among other things. However, the construction of the Three Gorges Project has brought a series of impacts on the ecology and environment along the river belt and even the whole basin [88]. Landslide hazards are a widespread geological problem in the reservoir area, becoming the biggest threat and obstacle to the economic and social development of the reservoir area and a critical area for the protection and control of geological risks in China.

There are numerous studies on the Three Gorges Reservoir area in the current research. Nonetheless, most of them concentrate on the local area of the region, with only a few studies covering the entire Three Gorges Reservoir area. In this study, the three most used machine learning approaches have a predictive influence on the future occurrence of landslides in the Three Gorges Reservoir area. At the same time, the relative importance of the elements reveals that in the Three Gorges Reservoir, the distance from the river has the greatest influence, which is also consistent with the ecological environment characteristics of the area. The construction of the Three Gorges dam project has a more significant impact on the geological problems in the area. Meanwhile, there are many other predisposing factors of geological hazards in the Three Gorges Reservoir area [85,89], but in the research process of this paper, the nine screening index components have a high degree of relative independence and are dependent on the region's landslide susceptibility assessment prediction.

## 6. Conclusions

The Three Gorges Reservoir is a new reservoir project in China with strict geological disaster prevention and control system in the place. However, some of the reservoir area's geological disaster monitoring and warning points need our attention due to disasters and dangerous situations. It is critical to understand the landslide-prone areas in the region to reduce the impact of landslide hazard events. Aiming at 9539 landslide events across the Three Gorges Reservoir, this paper first used the Pearson correlation coefficient to filter the index factors before selecting nine index factors: elevation, slope, plan curvature, profile curvature, distance to fault, lithology, topographic wetness index, NDVI, and distance to

river. The landslide susceptibility assessment model system was then built using the logistic regression (LR), random forest (RF), and support vector machine (SVM) algorithm models, and the study area's landslide susceptibility distribution map was established. Precision, recall, F1-score, and overall accuracy were used to assess and compare the performance of the models. ROC curves and AUC values were used to assess the prediction accuracy and efficiency of the models. The results imply that all three models adapted in this study for partitioning landslide susceptibility are effective but the SVM model outperforms the others. The model's weights for each indicator element were assigned to distinguish the impact of the indicator factors on landslides in the Three Gorges Reservoir area. According to the landslide susceptibility distribution map, the very high- and high-susceptibility areas for landslide occurrence are primarily distributed on both sides of the water system and in areas with significant changes in slope, and the results are consistent with the distribution of landslide sites. However, during the research process, it was discovered that the factors influencing the occurrence of landslides are not distinguished. In conclusion, the current research findings are beneficial not only to the current analysis field but also to other areas with similar topographic features and natural environments, as well as to the construction and development of the Three Gorges Reservoir area and the Three Gorges Reservoir project, as well as having a reference value for reducing the risk of landslides in the reservoir area and for land resource management, and this study will gradually increase in scope.

**Author Contributions:** Conceptualization, J.C., X.D. and J.L.; funding acquisition, J.L. and W.L.; methodology, J.C., X.D. and J.L.; software, J.C. and G.Q.; validation, J.C., X.D., W.L. and J.S.; resources, X.D., J.L., W.L., J.S. and Y.W.; data curation, J.C., G.Q., W.L. and Y.W.; writing—original draft preparation, J.C. and X.D.; writing—review and editing, J.C., X.D., Z.W. and J.L.; visualization, J.C., Z.W. and G.Q. All authors have read and agreed to the published version of the manuscript.

**Funding:** This research was funded by the National Key Research and Development Program of China (Grant No. 2021YFC3000401) and supported by the Open Foundation of the Research Center for Human Geography of Tibetan Plateau and Its Eastern Slope (Chengdu University of Technology) (NO. RWDL2021-ZD003); Key Research Bases of Humanities and Social Sciences in Higher Education in Sichuan Province, Sichuan Center for Disaster Economic Research (NO. ZHJJ2021-ZD001); and the National Natural Science Foundation of China under Grant 41671448.

**Data Availability Statement:** Not applicable.

**Acknowledgments:** We acknowledge any support given which is not covered by the author contributions or funding sections. Includes administrative and technical support that is not covered, as well as data materials for experiments.

**Conflicts of Interest:** The authors declare no conflict of interest.

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
