# Peer review of "Landslide Susceptibility Assessment Model Construction Using Typical Machine Learning for the Three Gorges Reservoir Area in China"

_remotesensing, doi:10.3390/rs14092257_

Round 1

Reviewer 1 Report

Based on the results shown by the submitted manuscript, it could be considered for publication if should you be prepared to incorporate minor revisions.

The manuscript deals with an interesting case study about Landslide Hazard Susceptibility Assessment Model Construction, and the focus of the study is clear enough from the beginning.

Even if dealing with a potentially very interesting topic and its general good writing in English, in the opinion of this review the manuscript should be reviewed by described following.

  1. The paper is structured sufficiently and clearly. I think it could be better to frame it in this way: Introduction, materials and methods, results and discussion, and conclusion.
  2. Generally, the introduction part needs to be refined, by adding some specific recent references in a broader context of the international literature available on the topic (international relevance should enhance).
  3. Try to improve the results part of your results to better explain the procedure for future applications.

Author Response

Comments and Suggestions for Authors

Comment 1: The paper is structured sufficiently and clearly. I think it could be better to frame it in this way: Introduction, materials and methods, results and discussion, and conclusion.

Response: Thank you for your valuable suggestions, we will check the article carefully and adjust the article structure accordingly.

Comment 2: Generally, the introduction part needs to be refined, by adding some specific recent references in a broader context of the international literature available on the topic (international relevance should enhance).

Response: We appreciate the comments and thank you pointing this out. We have carefully read the text and revised this question. At the same time, some specific latest references are added, and the revised content in the revised manuscript.

Comment 3: Try to improve the results part of your results to better explain the procedure for future applications.

Response: Thank you for pointing this out, we will carefully check the full text and improve the results section. And we have revised and improved as follows:

The Three Gorges Reservoir is a new reservoir project in China with severe geological disaster prevention and control. Some of the reservoir area's geological disaster monitoring and warning points need our attention due to disasters and dangerous situations. It is critical to understand the landslide-prone areas in the region to reduce the impact of landslide hazard events. Aiming at 9539 landslide events across the Three Gorges Reservoir, this paper first uses the Pearson correlation coefficient to filter the index factors before selecting nine index factors: elevation, slope, plan curvature, profile curvature, distance to fault, lithology, topographic wetness index, NDVI, distance to river. The landslide susceptibility assessment model system was then built using the logistic regression (LR), random forest (RF), and support vector machine (SVM) algorithm models, and the study area's landslide susceptibility distribution map was established. Precision, recall, F1-score, and overall accuracy are used to assess and compare the performance of the models. ROC curves and AUC values are used to assess the prediction accuracy and efficiency of the models. The results show that all three models used in this study for partitioning landslide susceptibility are effective, but the SVM model outperforms the others. The model's weights for each indicator element were used to determine the impact of the indicator factors on landslides in the Three Gorges Reservoir area. According to the landslide susceptibility distribution map, the very high and high susceptibility areas for landslide occurrence are primarily distributed on both sides of the water system and in areas with significant changes in slope, and the results are consistent with the distribution of landslide sites. However, during the research process, it was discovered that the factors influencing the occurrence of landslides are uncertain. In conclusion, the current research findings are beneficial not only to the current analysis field, but also to other areas with similar topographic features and natural environments, as well as to the construction and development of the Three Gorges Reservoir area and the Three Gorges Reservoir project, as well as having reference value to reduce the risk of landslides in the reservoir area and to land resource management, and this study will gradually increase in scope.

Reviewer 2 Report

Please consider the attached review report and revise it accordingly.

Regards

Author Response

Comments and Suggestions for Authors

Comment 1: “Landslides hazard susceptibility” is not correct term. It must be “landslide susceptibility”.

Response: Thank you for pointing this out, we will carefully check the full text and correct the statement error.

Comment 2: Line 67: the following studies should be considered for Random forest:

https://www.mdpi.com/2220-9964/9/9/553

https://www.frontiersin.org/articles/10.3389/feart.2021.712240/full

https://link.springer.com/article/10.1007/s12665-022-10225-y

https://www.tandfonline.com/doi/full/10.1080/10106049.2021.1996637

In addition, several different approaches and algorithms have been used when preparing landslide susceptibility maps. For example, fuzzy, neuro-fuzzy, and XBoost algorithms have been used. The authors can mention these methods.

Response: Thank you for your suggestions. We checked the context and added some relation algorithms references in the main text.

Comment 3: Lines 68-105: This part must be re-written completely or, remove.

Response: We appreciate the comments and thank you for pointing this out. We have carefully read the text and revised this question. The re-written sentence in the revised manuscript

Comment 4: Figure 1 is not completed. The search term “machine learning” is not sufficient. Some authors never used “machine learning” term. Some of the authors preferred to use the name of the algorithm directly. For this reason, this search must be performed again considering this comment.

Response: We appreciate the comments and thank you for pointing this out. We have carefully read the text and revised this question. The revised sentence is as follows:

Machine learning has gradually become the core body of artificial intelligence re-search and one of the fastest-growing disciplines of artificial intelligence with a wide range of applications, thanks to the rapid development of computer technology. The earliest machine learning algorithms date back to the early 20th century. After decades of progress, a large number of classical methods were born. Machine learning is now widely used in geological catastrophe research. The analysis of results of "machine learning" and "landslide" or "algorithm" and "landslide" through "Web of Science" shows that since 2010, there are 2658 types of research on landslide hazards and machine learning or algorithms by domestic and foreign scholars. The result of "machine learning" and "landslide susceptibility" or "algorithm "and "landslide susceptibility" is 1241 items, as shown in Figure 1.

Comment 5: Study Area section is meaningless, and this section must be re-written considering geological, geomorphological and hydrological properties of the area because these are directly related to landslide.

Response: Thank you for your suggestions. Based on the geographical location of the study area, we have added relevant content about the geological, geomorphological, hydrological and human engineering activities of the area, and rewritten this section of the study area. And we have revised and added a sentence as follows:

The Three Gorges Reservoir Dam on the Yangtze River, an important water conservancy project in China, and the Three Gorges Reservoir area is a significant ecological barrier in China, provide abundant water for irrigation in the Yangtze River basin, and have a considerable role in economic prosperity along the Yangtze River, promoting the economic development of the western region and balancing the East-West differences. The Three Gorges Reservoir region extends between 28°30′N and 31°45′N latitudes and 105°50′E and 111°42′E longitudes. It is connected with the Sichuan Basin, covering four districts and counties under the jurisdiction of Yichang City in Hubei Province and 22 districts and counties under the jurisdiction of Chongqing City, with a total area of about 79,000 square kilometers and submerged arable land of 19,400 hectares. The Three Gorges Reservoir area ecosystem is unique and fragile, with frequent natural disasters. At the same time, the Three Gorges Reservoir area is an ecological treasure trove for the Yangtze River Economic Belt and the whole country.

The Three Gorges Reservoir area of the Yangtze River is located in the mid-latitude subtropical monsoon climate zone, influenced by alternating winter and summer winds, the temperature and precipitation change significantly in seasons. The climatic characteristics are very obvious due to the complex terrain. According to the annual monitoring data, the less precipitation along the river valley in the study area, and the average annual rainfall increases by about 55 mm for every 100 meters in-crease in elevation. The rainy season is from May to September every year, and its rainfall accounts for 70% and more of the year, and there are many heavy rainstorms. The climate of Three Gorges reservoir area with abundant rainfall and heavy rainfall is one of the main triggering factors for the occurrence of landslide geological disasters in the reservoir area.

The study area is located in the transition zone from the second to the third terrace of China’s topography, and is the junction of the east Sichuan fold and the west Hubei mountains, with a middle and low mountain erosion canyon landscape. The east-west part of the reservoir area traverses two natural geographic units, roughly bounded by Fengjie, with the eastern part is the Three Gorges Canyon deeply embedded in the Wushan Mountains and the western part is the low mountainous hilly area in the eastern part of the Sichuan Basin. The loose rock pore water in the study area is mainly stored in the loose accumulation layer and slope accumulation layer of the Quaternary system, and is mainly recharged by precipitation, fracture water of the underlying bedrock or karst water, so it is influenced by seasonal changes. The dynamic instability of groundwater level is one of the main factors affecting the stability of landslide in the area.

Geological hazards in the study area are widely distributed, numerous, large scale and serious. Landslides are the most prominent type of geological hazards in the reservoir area, with a large number of developments, large scale and strong hazards. At the same time, with the rapid development of social economy, the scale and intensity of human engineering activities in this area have continued to expand, and the impact on the natural environment has become increasingly serious, which has become one of the important triggering factors of geological hazards in the area, mainly in the construction of migrant towns, reservoir construction, deforestation, mining and so on. These activities adversely affect the rock and soil bodies near the slopes, destroy the natural ecosystem, cause soil erosion, and seriously damage the original natural morphological structure and stress balance, which are important causes of landslides and collapse disasters.

Comment 6: Line 165: Which distribution? What do you mean?

Response: Thank you for pointing this out, we will carefully check the Figure 2 and adjusted the statement of the sentence.

Comment 7: Please describe landslides. What are the types of landslides? Which geological and geomorphological conditions control landslides in the study area? What are the main triggers etc?

Response: Thank you for pointing this out, we will carefully check the full text and correct the statement error. And we have revised and added a sentence as follows:

Landslide datasets are essential for investigating and analyzing regional landslide hazards and risks. Following reservoir storage, landslides, and debris flows have increased due to the complicated geological conditions and disasters in the Three Gorges reservoir area. Landslides in the Three Gorges Reservoir area mainly include accumulation layer landslides, bedding rock landslides, dangerous rock mass landslides, unstable slopes and reservoir banks. The external factors affecting the deformation of geological disasters in the reservoir area mainly include reservoir water, rainfall, and human engineering activities. In the initial stage of the Three Gorges Reservoir impoundment, reservoir water was the main inducing factor for the deformation of geological disasters in the Three Gorges Reservoir area. During the high water mark operation in recent years, rainfall become the dominant trigger. Under the action of different external forces, different levels of landslide disasters occur every year in this area. There are 9539 landslides in the landslide cataloging data, including 3661 large landslide events, 3852 medium landslide events, 1432 small landslide events, and 594 other types of landslide events. The spatial distribution of landslide sites in the Three Gorges reservoir area is shown in Figure 3.

Comment 8: Elevation or altitude?

Response: We appreciate the comments and thank you for pointing this out. We have carefully read the text and revised this question. We have read the article carefully, and the elevation mentioned in this article is also the altitude.

Comment 9: Precipitation is trigger or conditioning factor for the landslides in the study area?

Response: Thank you for pointing this out. Precipitation is one of the important triggering factors to induce landslides. Since the study area is located in the subtropical climate zone, with mild and humid climate, abundant and heavy or continuous rainfall, rainfall and other related effects are one of the main external forces of slope deformation and damage in the Three Gorges Reservoir area.

Comment 10: I do not understand the landslide-profile curvature relation? (figure 6d).

Response: We appreciate the comments and thank you for pointing this out. Figure 6d is a statistical graph of profile curvature and landslide density, from which it can be seen that the density of landslides is higher in the area where the profile curvature is between -0.03-0.03°/100m. Among them, the area between -0.03-0°/100m has the highest landslide density, which indicates that most landslides are concentrated in relatively gentle concave or convex areas. However, because the section curvature is gentle in the interval of 0~0.01°/100m and tends to be a straight slope; while the curvature changes more obviously in the interval from -0.03~0°/100m and 0.01~0.03°/100m for convex slopes and concave slopes, so the frequency of landslides is higher, and human activities and landslides form more accumulation materials here.

Comment 11: Is logistic regression a machine learning method or statistical method? Please be careful.

Response: Thank you for pointing this out, we will carefully check the full text and correct the statement error. And we have revised the main text as:

Logistic Regression (LR) is a standard linear regression analysis model for establishing the relationship between constraints and landslides [56; 57]. Regression problems based on categorical variables, dealing with linear relationships between numerical interval variables, based on a set of predictor variables that predicts the probability of an event occurring with a binary variable (such as 0 and 1 or true and false) [58]. The model's independent variables are independent factors, and the best fitting function is determined to quantitatively characterize the relationship between landslide occurrence and causes.

Comment 12: Conclusion section should be re-written highlighting the scientific findings obtained from the study.

Response: We appreciate the comments and thank you for pointing this out. We have carefully read the text and revised this question. The re-written sentence in the revised manuscript.

Reviewer 3 Report

Dear authors, you are presenting a high quality paper with very good results which can reduce landslides impact in a key region for the Chinese water dynamics.

My comments and corrections in the attached PDF.

All the best.

Author Response

Comments and Suggestions for Authors

Comment 1: Lines 43-44: check and add the following reference:

Worldwide Research Trends in Landslide Science. International Journal of Environmental Research and Public Health, 18(18), 9445. https://doi.org/10.3390/ijerph18189445

Response: Thank you for your suggestions. We checked the context and added your suggested references in the main text.

Comment 2: Line 45: here, check and add the following papers enhancing your argument:

Piciullo L, Calvello M, Cepeda J (2018) Territorial early warning systems for rainfall-induced landslides. Earth-Sci Rev 179:228-247. https://doi.org/10.1016/j.earscirev.2018.02.013

Segoni S, Piciullo L, Gariano SL (2018) A review of the recent literature on rainfall thresholds for landslide occurrence. Landslides 15:1483–1501. https://doi.org/10.1007/s10346-018-0966-4

Response: Thank you for your suggestions. We checked the context and added your suggested references in the main text.

Comment 3: Line 115-117: normally, the order of chronologies goes from left to right, please correct (Figure 1)

Response: We appreciate the comments and thank you for pointing this out. We have carefully read the text and revised this question. The revised sentence is as follows:

Machine learning has gradually become the core body of artificial intelligence research and one of the fastest-growing disciplines of artificial intelligence with a wide range of applications, thanks to the rapid development of computer technology. The earliest machine learning algorithms date back to the early 20th century. After decades of progress, a large number of classical methods were born. Machine learning is now widely used in geological catastrophe research. The analysis of results of "machine learning" and "landslide" or "algorithm" and "landslide" through "Web of Science" shows that since 2010, there are 2658 types of research on landslide hazards and machine learning or algorithms by domestic and foreign scholars. The result of "machine learning" and "landslide susceptibility" or "algorithm "and "landslide susceptibility" is 1241 items, as shown in Figure 1.

Comment 4: Line 164-165: please add extreme geographic coordinates of the three maps that comprise the figure (Figure 2)

Response: Thank you for pointing this out, we will carefully check the Figure 2 and adjusted the statement of the sentence.

Comment 5: Line 183-184: your maps are shown as islands, please put surrounding catchments or political units...in all maps (Figure 3)

Response: Thank you for pointing this out, we will carefully check the Figure 3 and added political units to all maps.

Comment 6: Line 192: your maps are shown as islands, please put surrounding catchments or political units...in all maps

Response: Thank you for pointing this out, we will carefully check the Figure 3 and added political units to all maps.

Comment 7: Line 243: your maps are shown as islands, please put surrounding catchments or political units...in all maps

Response: Thank you for pointing this out, we will carefully check the Figure 3 and added political units to all maps.

Comment 8: Line 470 Discussion: you have to discuss (1) your results with previous studies of landslides in the studied area, (2) similar studies and results in China and Asia, (3) studies in similar conditions worldwide

Response: Thank you for pointing this out. We have added some discussion and rewritten to Section 5.3 in the revised manuscript.

Comment 9: Line 486: please add photos of areas with high, medium, and low susceptibility...

Response: We appreciate the comments and thank you for your suggestion. We have carefully read the text and added relation photos in this part.

Comment 10: Line 531-532: how you justify the use of TWI and Slope in the same models? those variables have high collinearity...please explain

Response: We appreciate the comments and thank you pointing this out. We have carefully read the text and revised this question. In the process of selecting the indicator factors, we first preprocessed the data for this study area by terrain correction. Then, the Pearson correlation coefficient method and Collinearity Diagnostics screened out the nine correlation factors used in this paper, among which TWI and Slope in Collinearity Diagnostics process. Slope’s TOL is 0.302, VIF is 3.307; TWI’s TOL is 0.866, and VIF is 1.155. At the same time, the indicator factors that have strong collinearity problems with VIF≥5 or TOL≤0.2 are discarded from the model, which indicates that TWI and Slope are reasonable to use in our model.

Round 2

Reviewer 2 Report

The necessary revisions are penned and hence it can be accepted without further revision.

regards